# A study on turbulence characteristics of a rectangular three-dimensional wall jet in a confined space using particle image velocimetry

Xingxing Zhang[1,2]*, Han Liu[3]

1 School of Urban Construction Engineering, Chongqing University of Arts and Sciences, Chongqing, PR China, 2 National Engineering Research Center for Inland Waterway Regulation and Key Laboratory of Hydraulic and Waterway Engineering of the Ministry of Education, Chongqing Jiaotong University, Chongqing, PR China, 3 School of Civil Engineering, Chongqing Three Gorges University, Chongqing, PR China

* zhangxx@cqwu.edu.cn

## Abstract

A three-dimensional wall jet in a confined space is a ubiquitous flow occurrence in important engineering domains. Previous studies have not fully explained its turbulence characteristics. In this paper, a particle image velocimetry measuring technique was used to observe a rectangular three-dimensional confined wall jet. The objective of this study is to investigate specially turbulence statistics in regions affected by the vertical wall ($16 \leq x/d < 27$), and the effect of the Reynolds numbers and submerged depths on that. The results show that profiles of turbulence statistics are not similar in the region $16 \leq x/d < 19$. Turbulence characteristics in the region $19 \leq x/d < 24$ correspond to those of the radial decay region due to the influence of the vertical wall, where both the mean and fluctuating motions are completely developed in advance. In the region $24 \leq x/d < 27$, profiles of turbulence statistics in two planes collapse rather poorly. Peak values of turbulence quantities in the lateral plane show a positive correlation with Reynolds numbers obviously, while the similar trend is not presented in the symmetry plane. Furthermore, in the region $24 \leq x/d < 27$, profiles of turbulence intensities are little sensitive to small variations in submerged depths. However, peak values of Reynolds shear stresses show a positive correlation. This study can promote the understanding of turbulence characteristics of a three-dimensional confined wall jet.

## Introduction

A three-dimensional (3D) confined wall jet has numerous practical engineering applications because of its intriguing mean flow developments and turbulence characteristics. For example, a 3D wall jet in a confined space is widely used in local scour of riverbed [1,2], heat transfer [3,4], coastal wastewater treatment [5], spacecraft takeoff and landing [6], dam flooding [7] and also as prototypical flow in the filling and

**Data availability statement:** All relevant data are within the manuscript and its Supporting information files.

**Funding:** This research was funded by the Postdoctoral Science Foundation of China (2022MD713701), the Natural Science Foundation of Chongqing (CSTB2023NSCQ-MSX0860), the Science and Technology Research Program of Chongqing Municipal Education Commission (KJQN202201309), the Open Foundation of National Engineering Research Center for Inland Waterway Regulation and Key Laboratory of Hydraulic and Waterway Engineering of the Ministry of Education of Chongqing Jiaotong University (SLK2021A08), and Intelligent Construction and Health Operation for Mountain Projects Innovation Foundation of Chongqing University of Arts and Sciences (CXTD202308). The funders had no role in study design, data collection and analysis, decision to publish, or preparation of the manuscript.

**Competing interests:** The authors have declared that no competing interests exist.

emptying systems of navigation ship locks (Fig 1) for studying the momentum transport as well as energy dissipation [8]. During the filling of side orifices longitudinal culvert filling and emptying systems frequently employed in ship locks, the upstream flow with a certain initial momentum through side orifices injects into the chamber, resulting in the production of side orifices jets. Due to the limitations imposed by bottom plates, each side orifice jet shows flow characteristics of a 3D wall jet. Complicated boundary constraints that created a confined space also make the fluid flow in the chamber more intricate. On the one hand, lock walls and energy dissipation facilities generate vertical fixed boundaries that confine the overall growth of 3D wall jets, causing flow characteristics of multi-bound and non-free jets. Side orifices jets, on the other hand, have been in the submerged environment when filling, and the submerged depths change dynamically with time in a nonlinear manner, forming not only the horizontal active boundaries, but also flow characteristics of submerged jets. As a result, the effect of complex boundaries on the flow structure of a 3D confined wall jet has recently become a tough subject in fluid dynamics.

Experimental studies have proven to be an excellent way for researchers to investigate and comprehend the flow structure of a 3D wall jet. Using the concept of statistical averaging, turbulent flows may be split into mean and fluctuating motions since the fact that random variables have regular probability distributions and mean features. Fluctuating motion is vital for understanding shear layer formation, interaction between the inner and outer layers, and energy dissipation, whereas mean motion is a prerequisite and energy guarantee for fluctuating motion to occur. Therefore, considerable early experimental achievements observed mean developments of a 3D wall jet produced in a semi-infinite space by various shapes of nozzles (rectangular, square, circular, etc.). e.g., Sforza and Herbst [9], Newman et al. [10], Rajaratnam and Pani [11], Swamy and Bandyopadhyay [12], Narain [13], Davis and Winarto [14], Padmanabham and Gowda [15]. Subsequently, mean flow developments of the unconfined case were summarized in Law and Herlina [16] and Hall [17], including the devision of flow regions, profiles of mean velocities, spreading of velocity-half-widthand velocity-half-height, decay of local maximum velocity, etc. The 3D confined wall jet has also attracted a lot of research attention owing to various engineering applications associated with it, e.g., Onyshko et al. [18], Langer et al. [19], Chen et al. [20] and Zhang et al. [21]. Especially, Ref. [20] and Ref. [21] focused on showing mean characteristics with different Reynolds numbers and submerged depths in the regions affected by the vertical wall. These studies showed that the mean flow development of a confined jet remained unaffected by the vertical wall until $x/d = 16$, while the other regions were significantly characterized by the vertical wall. Fig 2 shows flow regions of a 3D wall jet in a semi-infinite and confined space. Mean flow characteristics in region II and region III were presented emphatically, including the definition of three regions along the streamwise direction, profiles of mean velocities, variations of velocity-half-height and velocity-half-width and decay of the maximum velocity. The influence of Reynolds numbers on mean characteristics in region II and region III was discussed in Ref. [20], while the influence of submerged depths in region II was studied significantly in Ref. [21]. It is noted that mean characteristics in

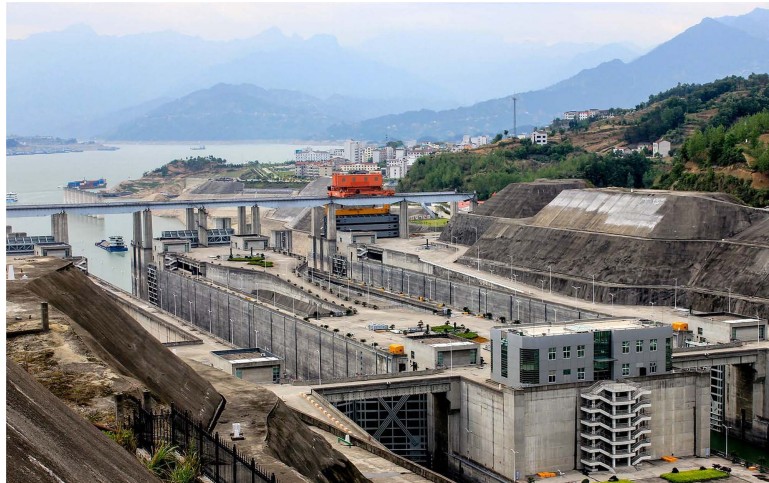

**Fig 1. Schematic diagram of ship locks in operation.**

region II reported by Ref. [20] and Ref. [21] were similar to those of the radial decay region. Therefore, the results showed that the vertical wall helped the 3D wall jet to develop. According to their investigations, the influence mechanism of the vertical wall for mean flow developments was set up, which is the research foundation for studying turbulence characteristics of a 3D confined wall jet. In summary, the preceding findings on mean flow developments of a 3D wall jet provide significant experimental information for determining turbulence characteristics.

However, few experimental studies concerning turbulence characteristics of the 3D confined wall jet have been documented attributed to the coupling influence of the vertical and bottom horizontal walls on the flow field. In the past, study findings concentrated on the semi-infinite space. Fujisawa and Shirai [22] investigated the flow field of a three-dimensional wall jet formed by a square orifice and discovered that the profiles of turbulence statistics in the radial decay (RD) region were self-similar and independent of the Reynolds number and nozzle shape. Karlsson et al. [23] used a laser doppler velocimetry technique to explore turbulence characteristics of a two-dimensional plan wall jet (Re = 10,000), focusing on the variations of Reynolds shear stresses and turbulence intensities in the boundary layer. Padmanabham and Gowda [24] assessed turbulence characteristics deeply within the RD region by getting the mean flow developments of a three-dimensional wall jet (Re = 95,400) [15] in different circular orifices. It was discovered that profiles of turbulence statistics were substantially self-similar in the symmetry plane further downstream from the orifice. In addition, they also found that the existence of the bottom wall boundary caused the 3D wall jet to have a higher turbulence level than that of the 3D free jet reported in Ref. [25]. Through circular pipe jet experiments, Abrahamsson et al. [26] and Sun and Ewing [27] confirmed this conclusion. According to a review of early investigations, hot wire anemometry was revealed to be the most commonly performed measurements. It is a single-point contact measurement with a high interference with the flow that cannot accurately measure the surface flow field in a short period of time. A particle image velocimetry (PIV) technique has steadily been applied to the measurement of a 3D wall jet as optical technology has advanced. Agelin-Chaab and Tachie [28] used the PIV technique to investigate turbulence characteristics of a 3D wall jet issuing from a circular orifice in a semi-infinite space (Re = 5,000,10,000, and 20,000). According to their findings, cases with high Reynolds numbers may produce a more intense entrainment of the ambient fluid, and profiles of turbulence statistics became self-similar at shorter distances. Kim et al. [29] studied turbulence characteristics of a 3D curved wall jet on a cylinder with varied incidence angles (Re = 3,300,7,100, and 11,800). Considering the Coanda effect and centrifugal force, the values of turbulence statistics in the symmetric plane were lack of collapse in their experimental results. Bode et al. [30] and Si et al. [31] will not be

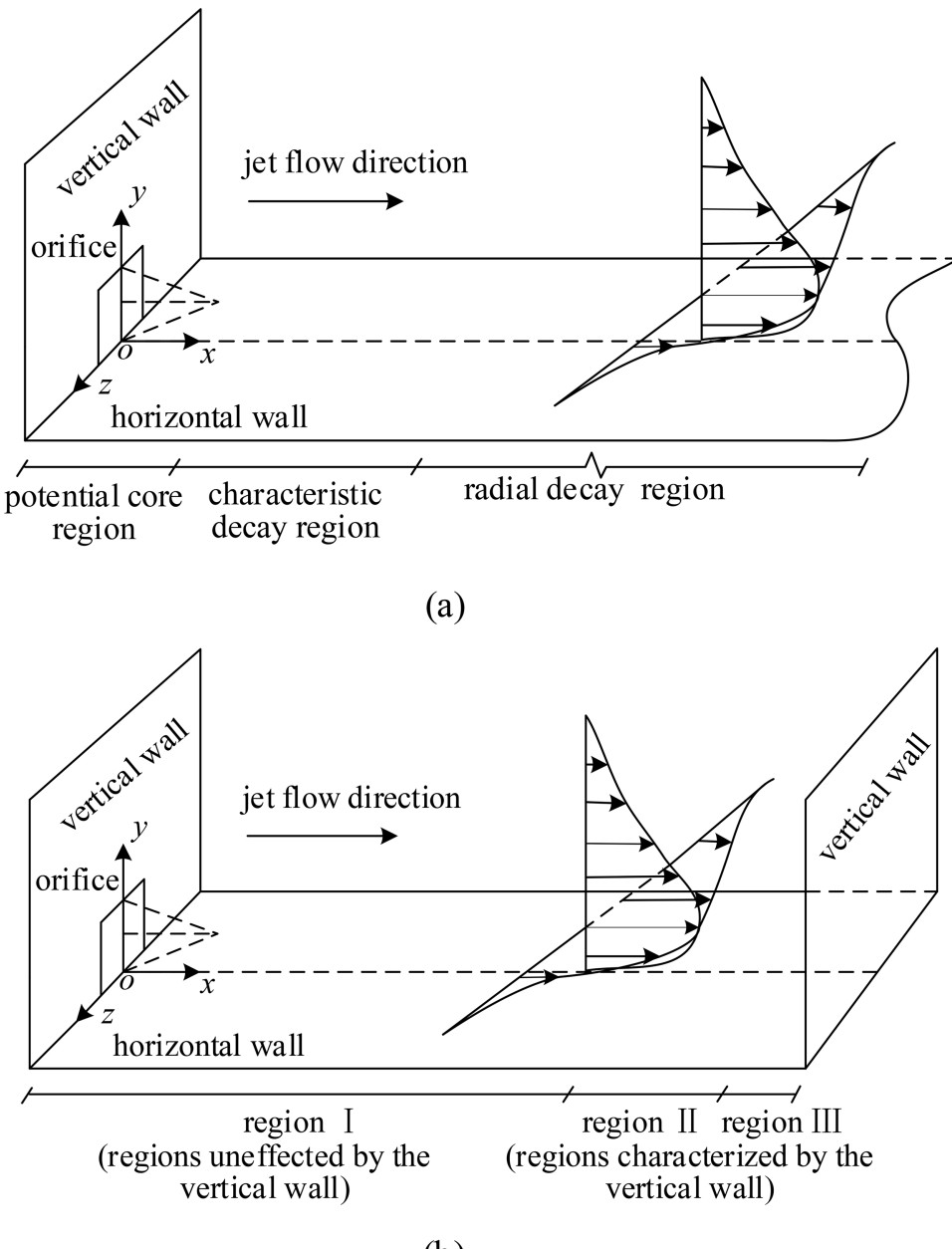

(a)

(b)

**Fig 2. A sketch of flow regions of a 3D wall jet in a semi-infinite and confined space, (a) a semi-infinite space (after Sforza and Herbst [9]) and (b) a confined space (after Zhang et al. [21]).**

reviewed here. The literature has already documented turbulence characteristics of the 3D wall jet in a semi-infinite space, including profiles of turbulence intensities and Reynolds shear stresses, but nothing is known about the confined case. Especially, the influence of the vertical wall and submerged depths is still unclear.

The goal of the present study is to investigate the effect of the vertical wall, Reynolds numbers and submerged depths on turbulence characteristics of a 3D confined wall jet, with velocity data obtained using PIV. Particularly, turbulence

intensities and Reynolds shear stresses in both the symmetry and lateral planes in the mean flow regions influenced by the vertical wall are reported and discussed in detail.

## Experimental setup and measurement system

### Confined wall jet facilities

The confined wall jet experiments were performed in a side orifice jet system, which primarily consisted of a test tank and supply-return facilities (Fig 3). A confined jet space is formed because of two lock walls which are defined as vertical walls in this study, based on the design manual of ship locks made by U.S. Army Corps of Engineers. The test section is the core component of confined wall jet facilities, which recreates a confined 3D wall jet as accurately as feasible when water filling in lock chambers. Therefore, the design of the width with the test section which must simulate exactly the confined

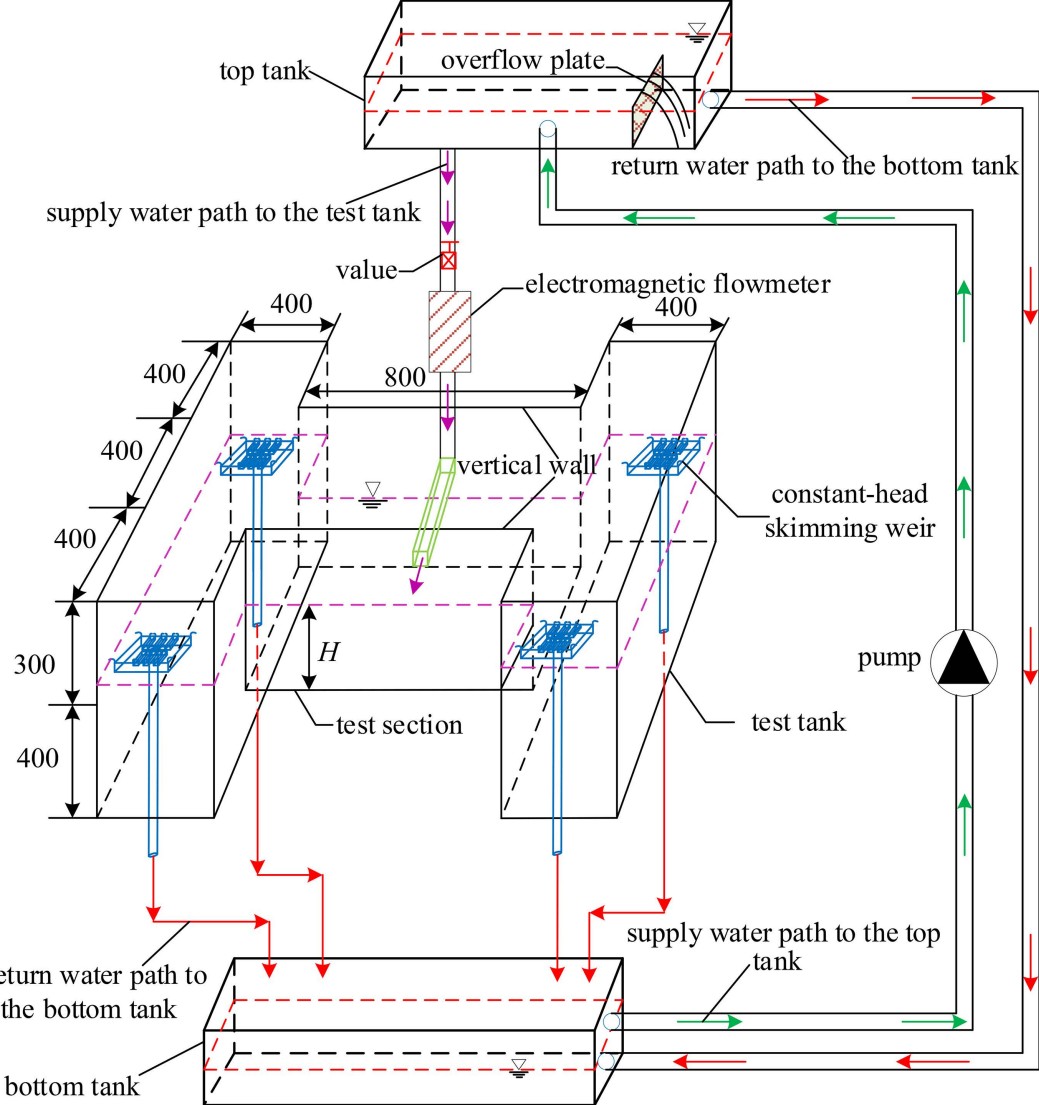

**Fig 3. A sketch of the confined wall jet facility (NB: All dimensions in mm).**

space is particularly important. A summary of typical ship locks with side orifices longitudinal culvert filling and emptying systems is provided in Table 1 [8]. It is shown that most real widths of lock chambers are within the range of 12m-25m, resulting in the fact that 20m is selected as the prototype size to build the width of the test section in this study. Considering the experimental field, range of laser irradiation, observation range of the CCD camera, accuracy requirements for images, etc., the test section was designed according to the normal model which should obey the similarity law of gravity, and its geometric scale was determined to be 1:50. i.e., $\lambda_l = \lambda_h = 50$, $\lambda_u = \lambda_l^{1/2} = 7.07$. Note that $\lambda_l$ represents the horizontal scale, $\lambda_h$ represents the vertical scale, and $\lambda_u$ represents the velocity scale, respectively. Its width was determined to be 400 mm by calculation. Due to difficulties in reconstructing the whole length of the lock chamber, the length of the test section was designed to be twice of the width to facilitate better diffusion of a 3D wall jet in the spanwise direction. Moreover, the test section was set as 300 mm high to provide greater activity space for constant-head skimming weirs. The side orifice was rectangular in shape with 14 mm wide as well as 16 mm high. To assist the incidence of the sheet light source, the test tank was made of flat glass with outstanding light transmission. A long rectangular pipe was linked to the bottom wall of the test tank via the glass side wall, forming a jet flow direction parallel to the centerline of bottom horizontal wall. A valve connected the pipe to the top tank, and an electromagnetic flowmeter monitored the incoming flow. It should be noted that four symmetrically organized constant-head skimming weirs and two water level probes controlled the water level in the test tank. The initial positions of four constant-head skimming weirs were located at a height of 300 mm above the top of the test tank (Fig 3). At the same time, the flexible pipes connected constant-head skimming weirs to the bottom tank, creating a path for overflow. Through the rise and fall of constant-head skimming weirs, experimental conditions for different water levels in the test section were achieved. The supply-return facilities consisted of a top tank, a bottom tank, and a pump. External water was pumped to the top tank, whose depth was measured by a water level probe when filling. To remain a constant water level of the top tank, excess water returned to the bottom tank through the overflow plate. The supply-return water paths were shown in Fig 3. Note that the aim of the supply-return facilities was to provide a stable incident flow rate for the test tank. The setting processes of the flow rate and water level for each test case are shown in Fig 4, using confined wall jet facilities.

Fig 5 shows a sketch of a 3D confined wall jet and also defines the coordinate system and notations utilized. The coordinate origin (o) is located at the center of the bottom of the orifice. $x$, $y$ and $z$ represent the streamwise, wall-normal and spanwise directions, respectively. $U$, $V$, and $W$ represent the mean velocities in each direction, while $u$, $v$, and $w$ denote the turbulence intensities. According to Fig 5, $U_m$ is local maximum streamwise mean velocity, $U_{0.5}$ is half of $U_m$, $y_m$ is the wall-normal location where $U_m$ occurs, $z_{0.5}$ and $y_{0.5}$ are the lateral and wall-normal locations where $U_{0.5}$ occurs. Also, note that the x-o-z plane at $y = y_m$ and x-o-y plane at $z = 0$ are the lateral plane and symmetry plane, respectively. The x-o-z

**Table 1. Typical ship locks with side orifices longitudinal culvert filling and emptying systems (after Zhang et al. [8]).**

| Serial number | Names of ship locks | Scales of chambers (Length and Width) (m) | Initial depths (m) | Designed heads (m) | Filling time (min) |
|---|---|---|---|---|---|
| 1 | Vallee | 204 × 33.5 | 3.96 | 6.7 | 9.9 |
| 2 | Jackson | 204 × 33.5 | 3.96 | 10.86 | 10.7 |
| 3 | Colombia | 154 × 25 | 4.27 | 7.65 | 9.8 |
| 4 | Eisen Howen | 244 × 24.4 | 9.45 | 15.7 | 8.4 |
| 5 | First line of Guiping | 186 × 23 | 3.5 | 9.96 | 8.0 |
| 6 | First line of Dayuandu | 180 × 23 | 3.0 | 11.2 | 8.7 |
| 7 | First line of Guigang | 190 × 23 | 3.5 | 14.5 | 8.3 |
| 8 | Wheeler | 122 × 18.3 | 3.96 | 16.4 | 12.0 |
| 9 | Naji | 190 × 12 | 3.5 | 13.91 | 8.0 |

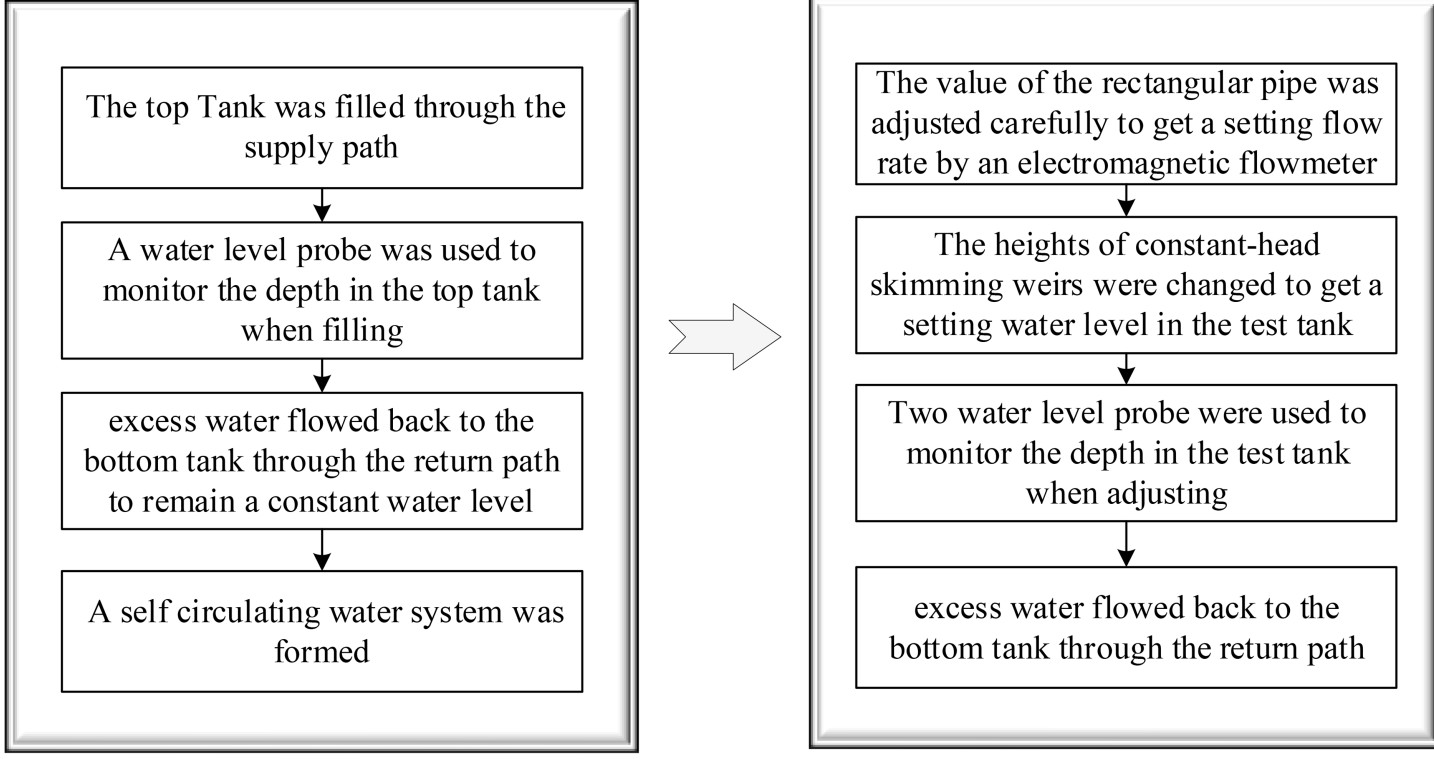

**Fig 4. The setting processes of the flow rate and water level for each test case.**

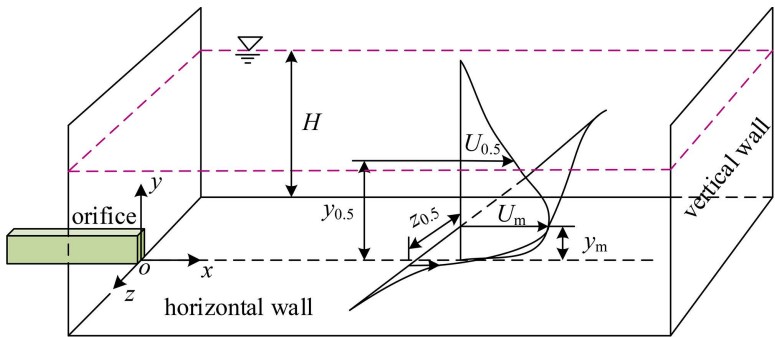

**Fig 5. Schematic diagram of a 3D confined wall jet.**

plane at $y=0$ is on the horizontal wall, the $y$-$o$-$z$ plane at $x=0$ is the orifice exit, and the $y$-$o$-$z$ plane at $x=400$ coincides with the vertical wall.

## PIV system and PIV algorithm

A PIV technique was used to performed the velocity measurements in the present study, including a laser, the CCD camera, a synchronization controller and powers. The flow was seeded with 10 $\mu m$ hollow glass spheres having a density of 1.03 g/cm³ to monitor the jet. A LWGL532 series laser with a 10W maximum power and a 532nm wavelength from Beijing

Laserwave Optoelectronic Technology Co., Ltd. was used to illuminate the flow field (Fig 6a). The light sheet entered the test section was 1 mm thick. An 8-bit high-resolution digital camera with a 2560 pixels × 1920 pixels charge-coupled device (CCD), an 800 Hz full frame frequency and a 4.8G memory was used to image the flow field, which was NX5-S2 series developed by IDT Corporation in the United States (Fig 6b). The CCD camera was equipped with a 50 mm lens (Canon 50 mm f/1.2). What is more, an optical filter with a 532nm wavelength matched the laser was applied in front of the CCD camera lens to significantly attenuate reflected stray light. To ensure all the illuminated images transmission, the camera talked with the server via an Ethernet based on the local area network. Note that the double-exposure mode which featured long-time and low-frequency was used to obtain all the instantaneous image pairs for the confined wall jet experiments, with a sampling frequency of 1 Hz and an exposure time of 400 μs. A time interval between two consecutive images was critically set to 1250 μs [20]. The particle pixel diameter was approximately 2.06 pixels for all the measurements, which was consistent with the value reported in Ref. [28]. Fig 7a and 7b show the arrangements of the laser and camera for measuring both the lateral and symmetry planes. It should be noted that the position of the x-o-z plane depends on where $U_m$ occurs. Considering that most $U_m$ in the RD region occur extremely on or close to the centerline of the orifice [16], the center plane of the orifice was focused on in the present study (i.e., the x-o-z plane at y = 8). The effect of a small offset in the wall-normal direction along the centerline of the orifice on the flow field in the lateral plane is ignored. In particular, a synchronization controller was also used to ensure that the camera and laser worked in tandem.

A Joy Fluid Measurement (JFM) software developed by Beijing Jiang Yi Technology Co., Ltd. was used to calculate the displacement increments of particles among instantaneous image pairs. The instantaneous flow field created by each instantaneous image pair would be determined through an instantaneous displacement and a time interval. JFM divided two consecutive images (i.e., image 1 and image 2) into numerous interrogation areas. The minimum interrogation window size was 16 pixels×16 pixels, and the horizontal and vertical grid overlap coefficient were set to 0.5 in JFM, so the final resolution was 8 pixels×8 pixels to improve the calculation accuracy. Images were reinterpreted by the multiple interpretation and an iterative multigrid image deformation method to avoid distortion. The number of iterations in the present calculation was set to two, which meant that each interrogation window in image 1 was cross-correlation calculated with one, two, and four times the size of search window in image 2, respectively. Horizontal and vertical offsets of windows were ignored. The fast Fourier transform method was used to realize the cross-correlation calculation between interrogation window pairs, and the three-point Gaussian curve fit was also employed to achieve subpixel interpolation on the correlation peak of the displacement [20,21]. Furthermore, after each iteration of the calculation, the normalized median test method reported in Ref. [32] was used to exclude erroneous velocity vectors and new vectors were re-interpolated by the Gaussian-weighted interpolation method. Finally, the instantaneous flow field was filtered to help decrease instability throughout the iteration processes and avoid computation result dispersion. The statistical mean approach was used

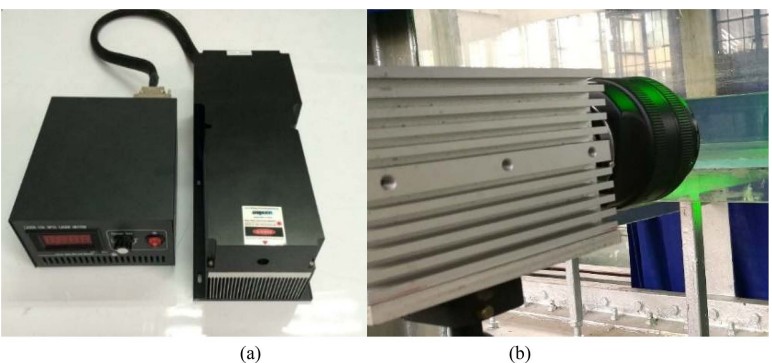

(a)                              (b)

**Fig 6. The laser and CCD camera, (a) a laser and (b) a CCD camera.**

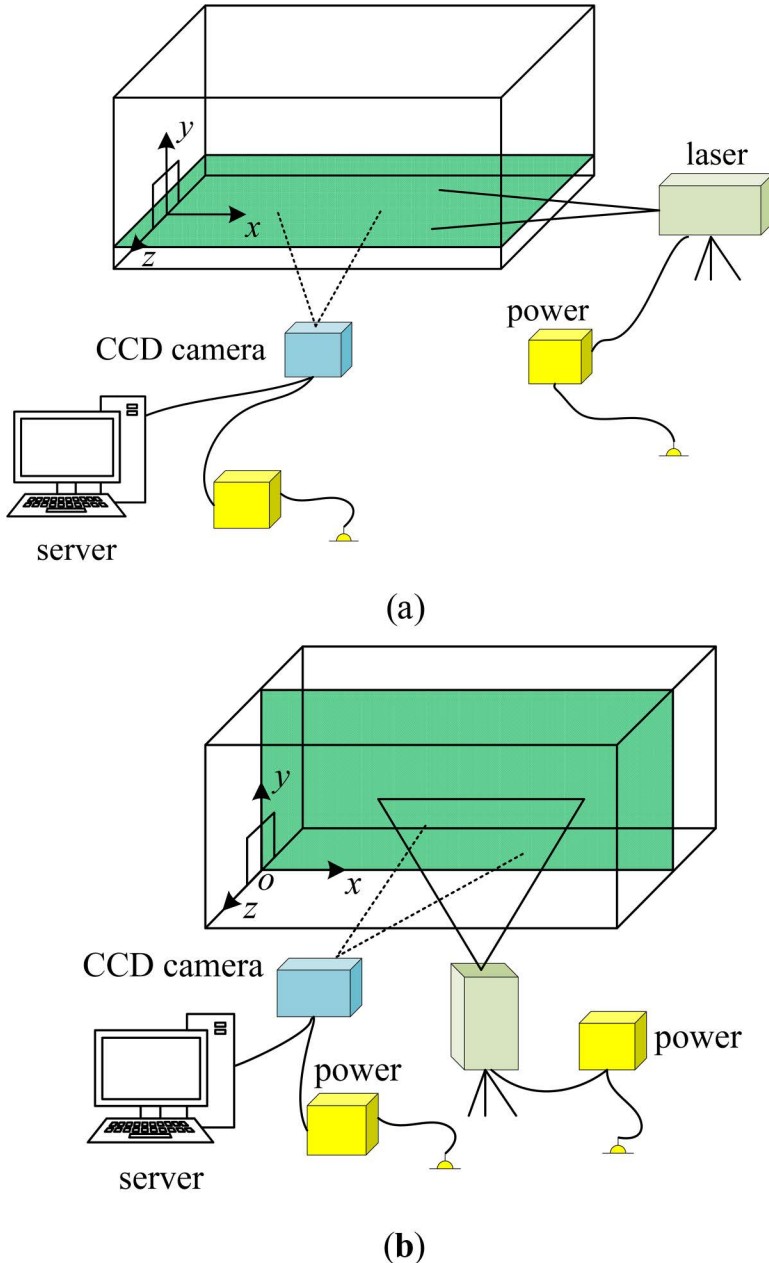

**Fig 7. Arrangements of the laser and camera, (a) the lateral plane setup and (b) the symmetry plane setup.**

to calculate the instantaneous velocity data acquired by JFM based on the turbulent stationary process and the ergodic theory, with the goal of determining the mean velocities, fluctuating velocities, turbulence intensities, and Reynolds shear stresses. The above calculation programs were compiled in our laboratory using MATLAB, and the number of instantaneous image pairs used to perform the statistical analysis was found to be 5000 pairs based on the mean velocities convergence tests in Ref. [20]. It should be noted that uncertainty analysis of the mean velocities had been documented in our earlier research results (Ref. [20]) and would not be discussed further here. In addition, considering the instantaneous

image pairs, interrogation window sizes and curve fitting algorithm, the uncertainties in the Reynolds shear stresses were estimated to be ±12.3% and ±10.5% close to and away from the wall, respectively. Notice that the uncertainties in this study were comparable to those reported in Ref. [28], ensuring the accuracy and availability of the analysis of subsequent turbulence characteristics.

## Test conditions and test cases

To limit local dispersion and intensity attenuation of the light sheet, the present PIV experiments were performed in a closed and shaded space of the laboratory. The test section was covered with a black cloth to prevent stray light from entering the camera. The initial temperature of the jet was 25°C, and the variation of the jet temperature was approximately 1°C to 2°C for all the measurements. As a result, temperature changes were neglected throughout velocity data processing. According to the initial depth in the chamber reported in Ref. [8], a minimum depth of the test tank was proposed to be 80 mm by the length scale conversion, and the submerged depths ($H$) increased by 0.25 times, which were 100 mm, 120 mm, and 140 mm, respectively. Similarly, though the velocity scale conversion, the jet exit velocities ($U_0$) were adjusted to 0.5 m/s, 0.6 m/s, and 0.7m/s based on the outflow velocities of side orifices [8], the associated Reynolds numbers were 8,333, 10,000, and 11,666 ($Re = U_0 d / v$, $d$ is the area equivalent diameter, i.e., square root of the jet exit area, and $v$ is kinetic viscosity of water at 25°C), and the associated flow rates were 0.0891L/s, 0.1091 L/s, and 0.1311 L/s, respectively. Besides, the levels of turbulence intensities at the inlet were 0.0518, 0.0506, and 0.0496 which corresponded to Reynolds numbers of 8,333, 10,000 and 11,666, respectively. Note that Reynolds numbers of the present study were the same as those of experiments on mean flow developments in a 3D confined wall jet as shown in Ref. [20]. Table 2 shows conditions of 24 test cases. Besides, when the water level in the top tank remained constant, the pipe valve was gently opened to measure the flow rates by viewing the electromagnetic flowmeter. Subsequently, the height of constant-head skimming weirs was adjusted to facilitate settings of the submerged depths, and the stability of the water level in the test section was observed with two water level probes. While both the incident flow rate and the water level in the test tank remained constant for 0.5 hours, the measurement of each case may be performed.

## Results and discussion

Previous studies of mean flow developments in the test tank showed that a 3D confined wall jet may be divided into three regions along the streamwise direction on the basis of decay characteristics of local maximum velocity in Ref. [20] and Ref. [21].

(1) Region I ($0 \leq x/d < 16$): the jet is unaffected by the vertical wall, and it is regarded as a purely 3D wall jet within the region;

**Table 2. Test conditions and test cases.**

| case numbers | $H$/mm | Reynolds numbers | measurement planes |
|---|---|---|---|
| case S1~case S3 | 80 | 8333,10,000,11,666 | the symmetry plane |
| case S4~case S6 | 100 | | |
| case S7~case S9 | 120 | | |
| case S10~case S12 | 140 | | |
| case L1~case L3 | 80 | 8333,10,000,11,666 | the lateral plane |
| case L4~case L6 | 100 | | |
| case L7~case L9 | 120 | | |
| case L10~case L12 | 140 | | |

(2) Region II ($16 \le x/d < 24$): the vertical wall promotes jet development with mean flow characteristics similar to the RD region;

(3) Region III ($24 \le x/d < 27$): the vertical wall significantly confines a 3D wall jet, showing a corner jet and an upward wall jet features when colliding with it.

To be sure, the effect of flow rates on the velocity profiles is very important while studying the turbulent characteristics of a 3D wall jet. The effect of flow rates on the velocity profiles with different Reynolds numbers had been discussed detailly in previous studies (Ref. [20] and Ref. [21]). In particular, the streamwise velocity profiles were well collapsed and consistent with Gaussian distribution in region II, which were independent of Reynolds numbers and submerged depths. However, after the wall jet entered region III, the similarity of $U$ profiles disappeared owing to the influence of the vertical wall. Therefore, in this section, turbulence intensities and Reynolds shear stresses are discussed emphatically in the region II and region III based on past studies in our laboratory and compared with previous turbulence characteristics in the RD region of the turbulent 3D and 2D wall jet. Moreover, the effect of the vertical wall, Reynolds numbers and submerged depths on turbulence statistics is also discussed and reported.

**Turbulence statistics in region II**

**Similarity consideration.** Fig 8 shows profiles of turbulence statistics from cases S1 and L1 both in the symmetry and lateral planes within region II. Test conditions for two cases were as follows: Re = 8,333, $H$ = 80 mm. The statistical profiles are $16d \sim 23d$. The streamwise ($u$) turbulence intensities, wall-normal ($v$) turbulence intensities and Reynolds shear stresses ($uv$) in the symmetry plane are as shown in Fig 8a, 8c and 8e. In general, the measured values of $u$, $v$, and $uv$ have an increasing trend along the streamwise direction in the region $16 \le x/d < 19$. Comparing $19d, 20d$, $21d$, $22d$, $23d$ profiles, it appears that the growth trend has disappeared, and profiles of $u$, $v$, and $uv$ show reasonable collapse. Note that change rules similar to those shown above are observed as well in case L1 (Fig 8b, 8d and 8f). This illustrates that turbulence statistics in region II require a relatively longer streamwise distance ($3d$) to become self-similar than mean characteristics both in the symmetry and lateral planes shown in Ref. [20] and Ref. [21]. This is to be predicted because turbulence kinetic energy of fluctuating motion is generated by mean motion, which means that the mean flow developments have to be self-similar before the turbulence quantities.

**Effect of Reynolds numbers and submerged depths.** As shown in Fig 8, profiles of turbulence statistics (cases S1 and L1) located in the region $19 \le x/d < 24$ are collapsed well in both planes. Therefore, in order to investigate the effect of Reynolds numbers and submerged depths on turbulence characteristics and compare further turbulent features with those in the RD region, $21d$ and $22d$ are selected as typical profiles in this study. Fig 9 shows profiles of turbulence intensities and Reynolds shear stresses from cases S1, S2, S3, L1, L2, and L3 at selected locations for different Reynolds numbers in the region $19 \le x/d < 24$. Test conditions in Fig 9 were that $H$ was 80 mm, and Reynolds numbers were 8,333, 10,000 and 11,666, respectively. Fig 10 shows same profiles of turbulence intensities and Reynolds shear stresses from cases S3, S6, S9, S12, L3, L6, L9, and L12 for different submerged depths in the region $19 \le x/d < 24$. Test conditions in Fig 10 were that Reynolds number was 11,666, and $H$ were 80 mm, 100 mm, 120 mm, and 140 mm, respectively. The results indicate that selected profiles of turbulence statistics display identical variation trends, regardless of Reynolds numbers or small variations in depths. In particular, it has been also discovered that the phenomenon of $u > v > w$ is commonly viewed in $21d$ and $22d$ profiles. This demonstrates that the three-dimensional wall jet for the present study fluctuates strongly in the streamwise direction. Comparing the streamwise intensities and Reynolds shear stresses both in cases S3 and L3, obviously, the 3D wall jet has a higher turbulence level in the symmetry plane. This is consistent with the findings of Padmanabham and Gowda [24], Sun and Ewing [27], and Agelin-Chaab and Tachie [28] on turbulence characteristics in the RD region of an unconfined 3D wall jet.

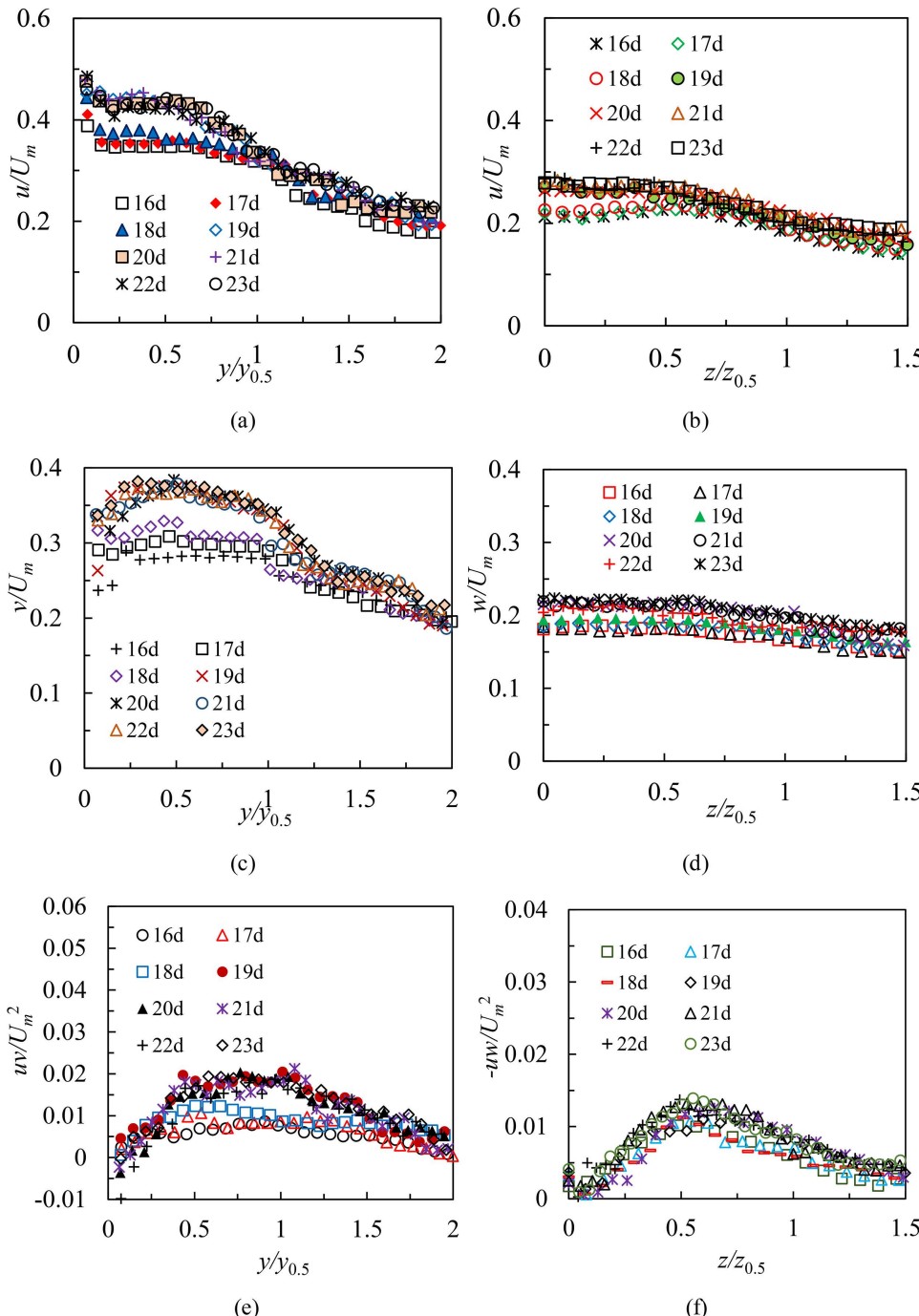

**Fig 8. Profiles of turbulence statistics from cases S1 and L1 at both in the symmetry and lateral planes within region II, (a) the streamwise (*u*) intensities in the symmetry plane, (b) the streamwise (*u*) intensities in the lateral plane, (c) the wall-normal (*v*) intensities in the symmetry plane, (d) the spanwise (*w*) intensities in the lateral plane, (e) Reynolds shear stresses (*uv*) in the symmetry plane and (f) Reynolds shear stresses (-*uw*) in the lateral plane.**

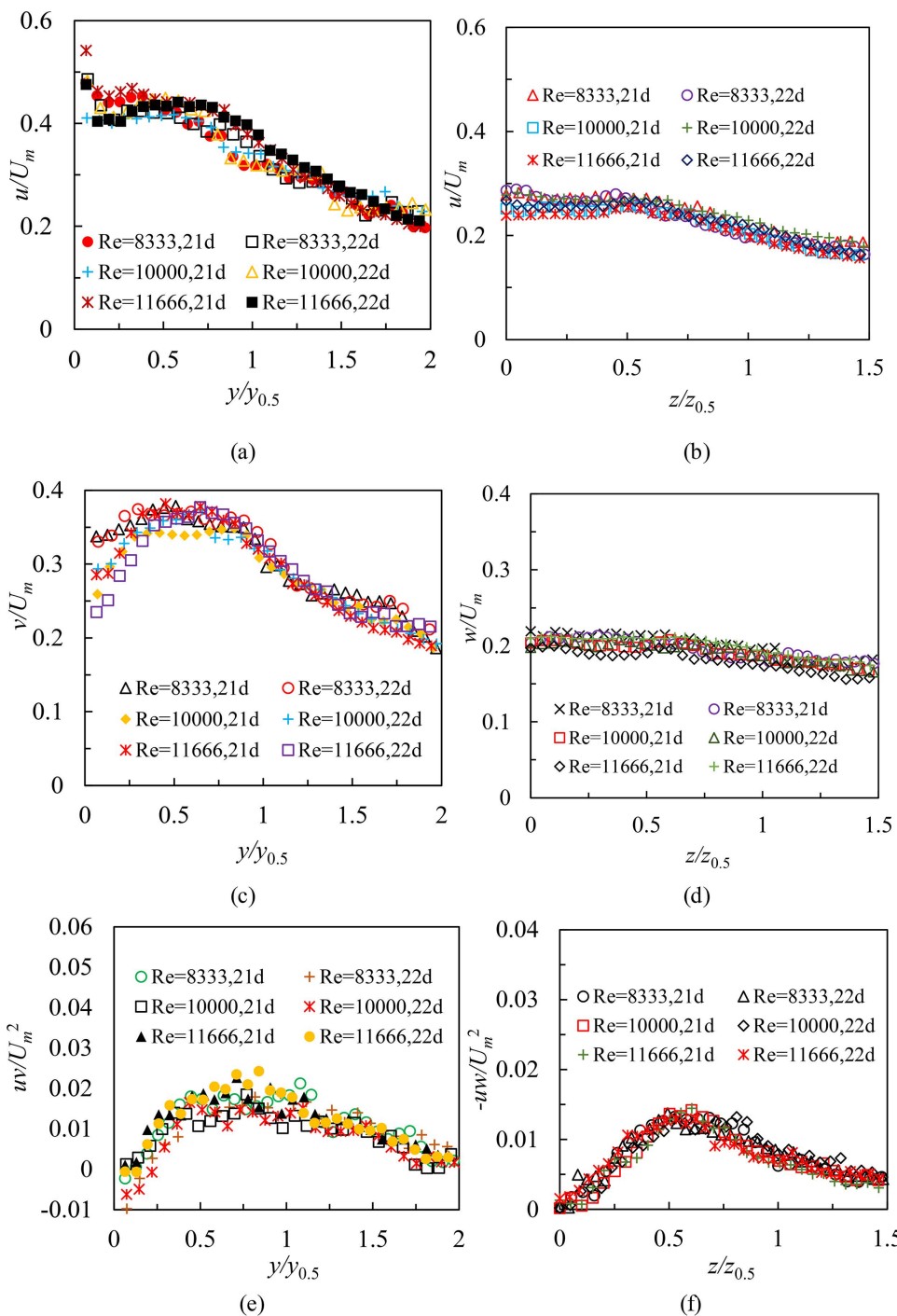

**Fig 9. Profiles of turbulence intensities and Reynolds shear stresses from cases S1, S2, S3, L1, L2, and L3 at 21d and 22d for different Reynolds numbers in the region 19 ≤ x/d < 24**, (a) *u* profiles for different Reynolds numbers in the symmetry plane, (b) *u* profiles for different Reynolds numbers in the lateral plane, (c) *v* profiles for different Reynolds numbers in the symmetry plane, (d) *w* profiles for different Reynolds numbers in the lateral plane, (e) *uv* profiles for different Reynolds numbers in the symmetry plane and (f) *-uw* profiles for different Reynolds numbers in the lateral plane.

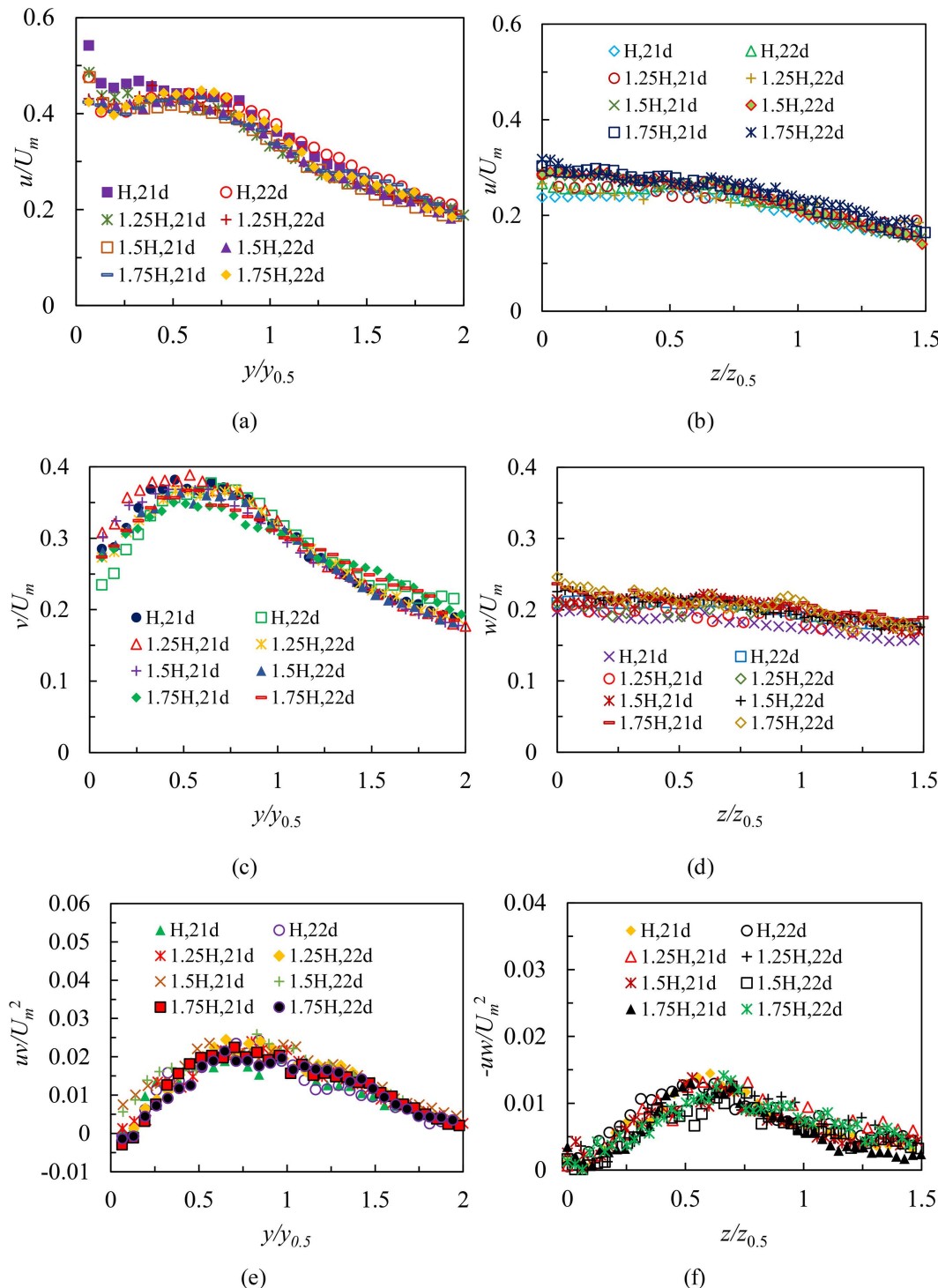

**Fig 10. Profiles of turbulence intensities and Reynolds shear stresses from cases S3, S6, S9, S12, L3, L6, L9, and L12 at 21*d* and 22*d* for different submerged depths in the region 19 ≤ *x*/*d* < 24, (a) *u* profiles for different submerged depths in the symmetry plane, (b) *u* profiles for different submerged depths in the lateral plane, (c) *v* profiles for different submerged depths in the symmetry plane, (d) *w* profiles for different submerged depths in the lateral plane, (e) *uv* profiles for different submerged depths in the symmetry plane and (f) -*uw* profiles for different submerged depths in the lateral plane.**

**Comparison and discussion with previous studies.** Fig 11 compares profiles of turbulence statistics from cases S3 and L3 at 22$d$ with the previous studies in the RD region (i.e., the location downstream of the orifice where $x/d = 40$, $x/d = 60$, or $x/d = 65$). As shown in Fig 11a, 11c and 11e, the peak values of $u$, $v$ and $uv$ all occur at $y/y_{0.5} > 0.5$. Considering that within region II, the location of maximum streamwise mean velocity ($U$) in the symmetry plane within region II corresponds to $y/y_{0.5} = 0.19$ [21], this demonstrates the maximum values of $u$, $v$ and $uv$ are located in the outer layer (i.e., $y > y_m$). The profile distribution of $u$ at 22$d$ is essentially the same as that of previous results in the RD region (Fig 11a). For example, the $u$ profile is very similar to that shown in Agelin-Chaab and Tachie [28] in the rigion $1.4 < y/y_{0.5} < 2$. While in the region $0.5 < y/y_{0.5} < 1.4$, the $u$ profile of the present study decays faster than those of other studies. In addition, the values of $u$ in the present study are much greater than the previous data sets. The peak value of $u$ is 17%, 21%, 69%, 65% and 57% higher than those reported by Agelin-Chaab and Tachie [28], Padmanabham and Gowda [24], Fujisawa and Shirai [22], Abrahamsson et al. [26], and Sun and Ewing [27], respectively. The present profile is also approximately twice of the 2D wall jet [23]. The peak value of $v$ as shown in Fig 11c is 45% higher than the data of Agelin-Chaab and Tachie [28], and Padmanabham and Gowda [24]. In addition, the maximum value of $v$ is twice of the previous data reported by Fujisawa and Shirai [22], Abrahamsson et al. [26], Sun and Ewing [27], and Karlsson et al. [23], respectively. This would imply that the present results in the symmetry plane obtain pretty higher turbulence intensities in the region $19 \le x/d < 24$. The reason for this is that the influence of the vertical wall significantly enhances the secondary flow motion, as more fluids are attracted by the bottom wall to form the motions pointing toward the bottom wall, which produce a more intense mixing with the jet and increase turbulence intensities in the symmetry plane. Fig 11e shows the $uv$ profile and the previous data sets. As can be observed, viscous stresses prevail extremely close to the wall and the values of $uv$ tend to zero, which may be attributed to the presence of the boundary layer. As soon as the jet leaves the bottom wall, the $uv$ profile continues to quickly grow, and the maximum value of $uv$ occurs at $y/y_{0.5} = 0.8$. Most importantly, it has discovered that the location where the value of $uv$ is zero is not consistent with that where $U$ is at its greatest in the symmetry plane. This observation would demonstrate that the phenomenon of counter-gradient for momentum transport occurs in the symmetry plane. This is a typical feature both in the 3D wall jet and 2D wall jet that had been documented by Agelin-Chaab and Tachie [28], Padmanabham and Gowda [24], and Karlsson et al. [23]. The location of $uv = 0$ corresponds to $y/y_{0.5} = 0.136$ in the experiments, which is consistent with the result of Padmanabham and Gowda [24] (i.e., $y/y_{0.5} = 0.13$). Besides, the present $uv$ data are comparable to the previous profiles, implying that that a pretty considerable measurement accuracy is present in the confined experiments. In summary, the peak values of turbulence quantities are closer to the bottom wall in the region $19 \le x/d < 24$ than in the 2D wall jet reported by Karlsson et al. [23] due to a higher interaction between the inner and outer layers.

Fig 11b, 11d and 11f show profiles of turbulence statistics at 22$d$ from case L3 and the previous profiles in the lateral plane. The $u$ profile in the lateral plane decays rather slight as the spanwise distance increases, which is consistent with the trend documented in previous studies (Fig 11b). For example, in the region $0 < z/z_{0.5} < 0.5$, the observed values of $u$ in the lateral plane are close to those of Sun and Ewing [27]. In the region $1 < z/z_{0.5} < 1.5$, the present data are in good agreement with those reported by Padmanabham and Gowda [24]. The maximum value of $u$ in the lateral plane occurs at the jet centerline, which corresponds to the data reported by Fujisawa and Shirai [22], Abrahamsson et al. [26], and Sun and Ewing [27]. But the present peak value of $u$ in the lateral plane is 36% and 19%, respectively, lower than those of Agelin-Chaab and Tachie [28], and Padmanabham and Gowda [24] due to their higher background turbulence levels in the lateral plane. The spanwise ($w$) turbulence intensity data compare favorably with those of Fujisawa and Shirai [22], and Wygnanski and Fiedler [25] in Fig 11d. The previous $w$ profiles reported by Agelin-Chaab and Tachie [28], Abrahamsson et al. [26], and Sun and Ewing [27] decay faster than the present profile. However, the measured values of $w$ coincide quite well with those of Agelin-Chaab and Tachie [28] in the region $1 < z/z_{0.5} < 1.5$. The value of Reynolds shear stress ($-uw$) in the lateral plane is zero at the jet centerline (Fig 11f), which corresponds to the location where the maximum $U$ in the lateral plane occurs [21]. Subsequently, the $-uw$ profile grows rapidly in the region $0 < z/z_{0.5} < 0.6$.

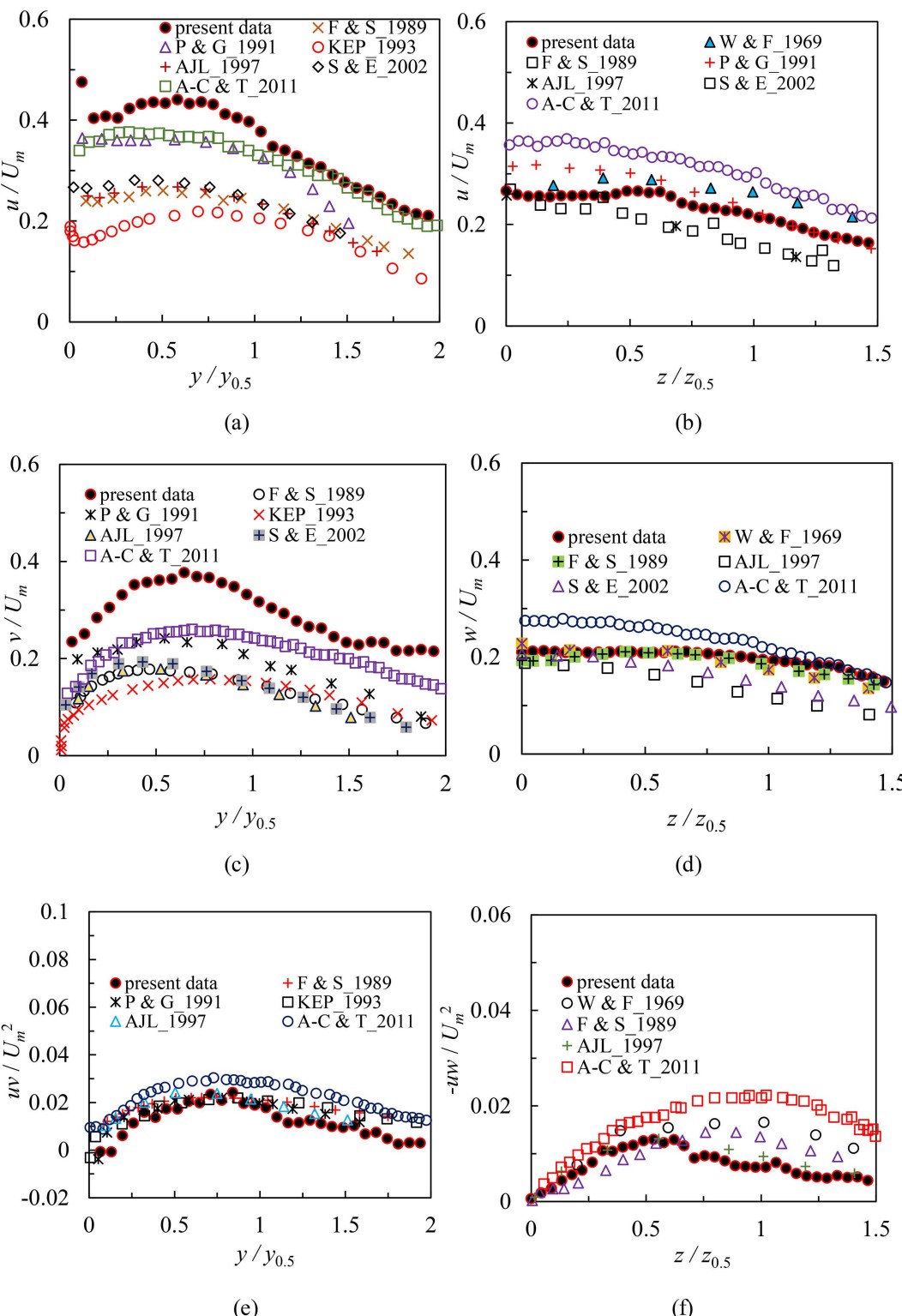

**Fig 11. Profiles of turbulence statistics from case S3 and case L3 at 22*d* in comparison with previous studies in the RD region, (a) *u* profiles of case S3 and previous *u* data in the symmetry plane, (b) *u* profile of case L3 and previous *u* data in the lateral plane, (c) *v* profile of case S3 and previous *v* data in the symmetry plane, (d) *w* profile of case L3 and previous *w* data in the lateral plane, (e) *uv* profile of case S3 and previous *uv* data in the symmetry plane and (f) -*uw* profile of case L3 and previous -*uw* data in the lateral plane.**

This would imply that the high-velocity fluids from the orifice continuously spread to the surrounding low-velocity fluids, resulting in a strong entrainment effect and a significant momentum exchange. Within the region $0.6 < z/z_{0.5} < 1.5$, the $-uw$ profile displays a declining trend, showing that the entrainment effect of the jet weakens as the mean velocity decreases. In general, a distribution trend similar to the $-uw$ profile in the present study is shown in the previous data sets in the lateral plane, as shown in Fig 11f.

Finally, turbulence characteristics in the region $19 \le x/d < 24$ correspond to those in the RD region of the 3D unconfined wall jet according to reviewing these important features of turbulence quantities, which are that the profiles collapse reasonably well, the profiles distributions are independent of Reynolds numbers and submerged depths and the present data are comparable to the previous studies. Combined with our previous studies reported in Ref. [20] and Ref. [21], both the mean and fluctuating motions are completely developed in the region $19 \le x/d < 24$. This demonstrates that it takes a much shorter streamwise distance to enter the region of flow structure similar to the RD region in this study than the unconfined case due to the help of the vertical wall.

## Turbulence statistics in region III

Fig 12 shows profiles of turbulence intensities and Reynolds shear stresses from cases S1, S2, S3, L1, L2, and L3 at selected locations for different Reynolds numbers in region III. The goal is to study the effect of Reynolds numbers on turbulence quantities in the region $24 \le x/d < 27$. As shown in Fig 12a, 12c and 12e, in terms of case S3, although turbulence statistics at $24d$ and $25d$ in the symmetry plane show the same distribution trend as those at $22d$, peak values of $u$, $v$, and $uv$ are much greater than the data at $22d$. The reason is that a stronger restriction by the vertical wall significantly enhances the turbulence level in region III. This would imply that profiles of turbulence statistics for case S3 do not collapse in the symmetry plane. However, as can be observed in Fig 12a and 12c, in the region $24 \le x/d < 26$, the $u$ and $v$ profiles of $24d$ and $25d$ for cases S1, S2, and S3 show an equal variation rule in the symmetry plane, and their values have little change with various Reynolds numbers. As the jet gets further closer to the vertical wall (i.e., $26 \le x/d < 27$), regarding case S3, turbulence intensities at $26d$ increase rapidly. This may be attributed to the change of flow characteristics. i.e., After the jet collides with the vertical wall, most fluids in the symmetry plane will be along the vertical wall to produce an upward wall jet, and the rest will form a backflow. A stronger influence of the backflow and the pressure gradient may be obtained close to the vertical wall, leading to relatively higher turbulence intensities. Fig 12e shows the $uv$ data at selected locations in region III. On the one hand, the qualities of collapse for the $uv$ profiles in region III are rather poor in comparison with the observations made in the region $19 \le x/d < 24$. On the other hand, peak values of $uv$ in region III show a positive correlation with Reynolds numbers by comparing $24d$ or $25d$ profile of cases S1, S2, and S3.

Regarding case L3, peak values of $u$, $w$, and $-uw$ at selected locations in the lateral plane within region III are also much larger than the observation measured at $22d$ in Fig 12b, 12d and 12f. The relatively higher turbulence statistics close to the vertical wall were got in the region $24 \le x/d < 27$ in comparison with those at $22d$ for case L3 may due to the formation of a corner jet. When the jet collides with the wall, it will proceed along two right-angle sides because of the Coanda effect to produce a corner jet, resulting in the fact that more fluids are entrained in the spanwise direction, and vortices of various sizes are developed near the vertical wall. This would enhance turbulence quantities in the lateral plane. In addition, a higher Reynolds number causes more ambient fluids mixing around. This would increase turbulence statistics in the lateral plane through a detailed analysis of $24d$ or $25d$ profile from cases L1, L2, and L3.

Fig 13 shows profiles of turbulence statistics from cases S3, S6, S9, S12, L3, L6, L9, and L12 at $25d$ for different submerged depths in region III. The objective is to explore the effect of depths on turbulence quantities. It is demonstrated that profiles of turbulence intensities both in the symmetry and lateral planes are not affected by small variations in depths within the region $24 \le x/d < 27$ in Fig 13a–13d. However, as shown in Fig 13e and 13f, a contrary law occurs in the profiles of Reynolds shear stresses both in the symmetry and lateral planes. The peak values of Reynolds shear stresses are positively correlated with depths, implying that the submerged depths promote the development of momentum transport.

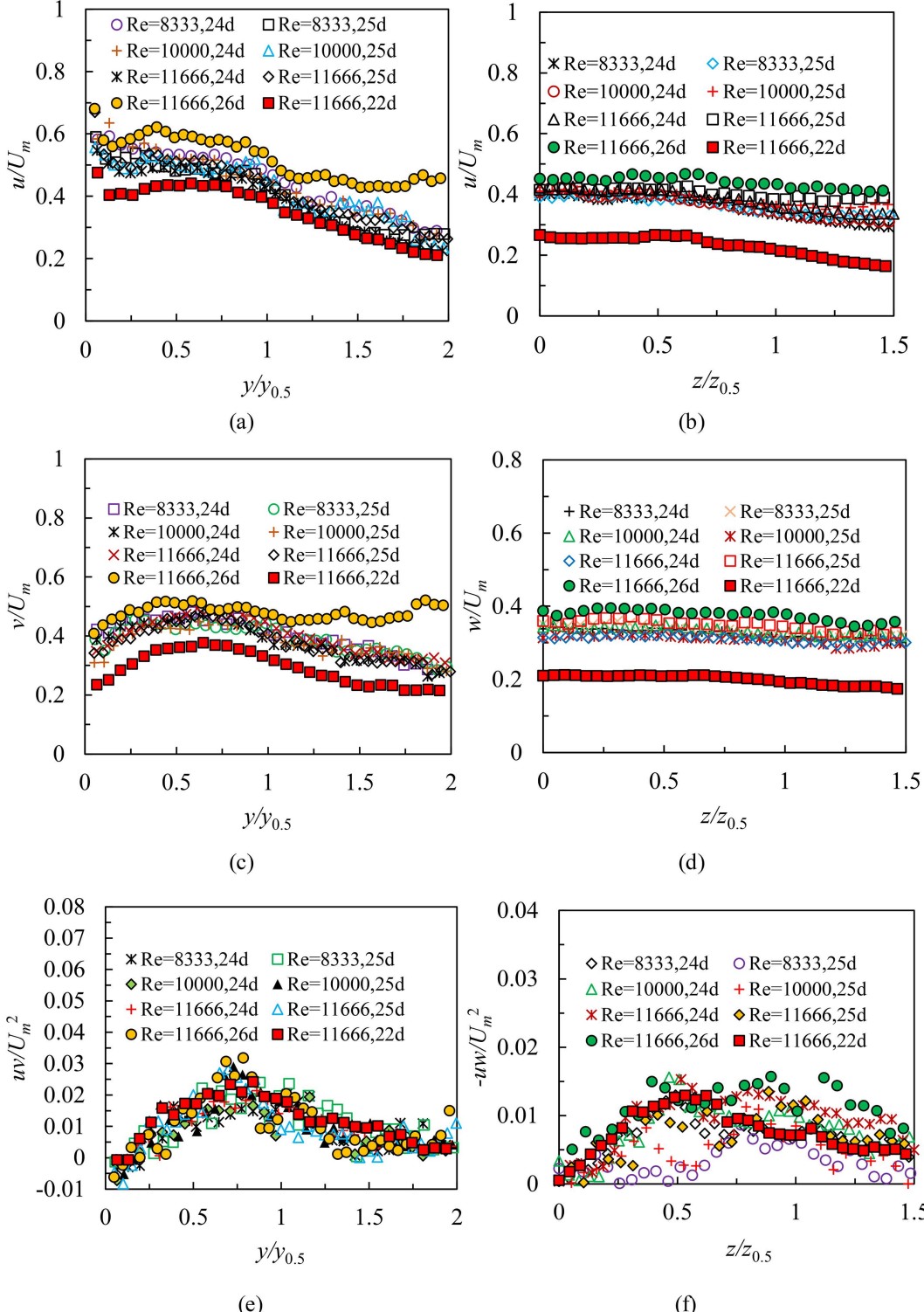

**Fig 12. Profiles of turbulence intensities and Reynolds shear stresses from cases S1, S2, S3, L1, L2, and L3 at selected locations for different Reynolds numbers in the region 24 ≤ x/d < 27, (a) u profiles at 24d, 25d for different Reynolds numbers and the data of 22d, 26d for case S3 in the symmetry plane, (b) u profiles at 24d, 25d for different Reynolds numbers and the data of 22d, 26d for case S3 in the lateral plane,**

**(c)** *v* profiles at 24*d*, 25*d* for different Reynolds numbers and the data of 22*d*, 26*d* for case S3 in the symmetry plane, **(d)** *w* profiles at 24*d*, 25*d* for different Reynolds numbers and the data of 22*d*, 26*d* for case S3 in the lateral plane, **(e)** *uv* profiles at 24*d*, 25*d* for different Reynolds numbers and the data of 22*d*, 26*d* for case S3 in the symmetry plane and **(f)** *-uw* profiles at 24*d*, 25*d* for different Reynolds numbers and the data of 22*d*, 26*d* for case S3 in the lateral plane.

What is more, profiles of Reynolds shear stresses in the lateral plane collapse even poorer as the increase of depths than those in the symmetry plane.

## Conclusions

The PIV technique was used to investigate turbulence characteristics of the 3D confined wall jet. The experiments were performed at three different Reynolds numbers of 8,333, 10,000, and 11,666, and four different submerged depths of 80 mm, 100 mm, 120 mm, and 140 mm. Focusing on the mean flow region influenced by the vertical wall ($16 \leq x/d < 27$), in addition to profiles of turbulence statistics, the effect of Reynolds numbers and submerged depths on turbulence quantities was applied to show turbulence characteristics in a confined space. The following findings were made as a result of this study:

(1) In the region $16 \leq x/d < 19$, turbulence intensities and Reynolds shear stresses increase along the streamwise direction, and the profiles show poor collapse. The qualities of collapse for turbulence statistics are rather reasonable in the region $19 \leq x/d < 24$, and their distributions are independent of Reynolds numbers and submerged depths. This demonstrates that turbulence statistics in region II take a longer streamwise distance ($3d$) to become self-similar than mean characteristics of a 3D confined wall jet both in the symmetry and lateral planes. The profiles distributions are consistent with the previous data sets of the RD region in comparison with previous studies. In terms of the salient features shown above, turbulence characteristics in the region $19 \leq x/d < 24$ correspond to those in the RD region of the unconfined case due to the influence of the vertical wall, where both the mean and fluctuating motions are completely developed in advance. This shows that the region of flow structure similar to the RD region in the 3D confined wall jet is further determined in this study.

(2) In the region $24 \leq x/d < 27$, profiles of turbulence statistics in the lateral plane do not collapse due to the vertical wall boundary effect which leads to the change of jet shape, and the peak values of turbulence quantities are positively related to Reynolds numbers and the distance from the vertical wall. The effect of the vertical wall on turbulent characteristics appears to be weaker in the symmetry plane within region III. Profiles of the streamwise and the wall-normal turbulence intensities in the symmetry plane show the equal trends in the region $24 \leq x/d < 26$, and their distributions have not much change with different Reynolds numbers. A similar trend is not observed in the region $26 \leq x/d < 27$ may be attributed to a stronger restriction by the vertical wall. However, profiles of Reynold shear stresses in the symmetry plane present the same distribution and change trends as those in the lateral plane in region III. Moreover, profiles of turbulence intensities are little sensitive to small variations in depths. On the contrary, the peak values of Reynolds shear stresses show a positive correlation with submerged depths. In particular, their profiles distributions in the lateral plane are more sensitive to variations in depths by comparing those with the symmetry plane.

(3) It is reasonable to draw conclusions that a higher turbulence level in a confined space can contribute to the faster momentum transport and energy dissipation than the unconfined case. According to research results, in terms of specific applications in real ship locks, a dissipation baffle similar to the vertical wall boundary is set up at a certain distance from the side orifice to form a confined jet space. Note that the distance and quantity of baffles need to be further determined according to the factors such as designed head, filling time and ship lock safe management, etc.

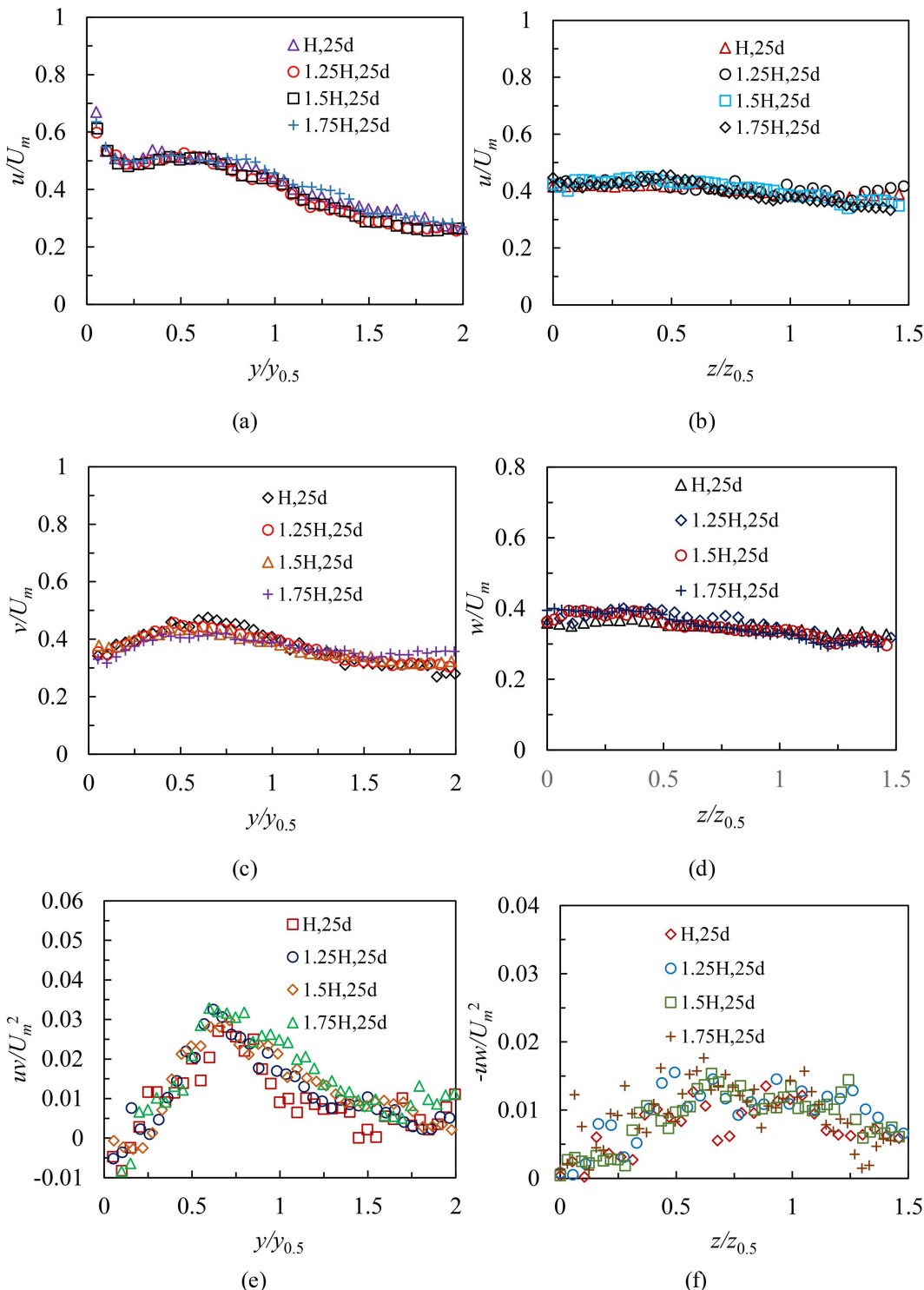

**Fig 13. Profiles of turbulence intensities and Reynolds shear stresses from cases S3, S6, S9, S12, L3, L6, L9, and L12 at 25$d$ for different submerged depths in the region 24 ≤ $x/d$ < 27, (a) $u$ profiles at 25$d$** for different submerged depths in the symmetry plane, (b) $u$ profiles at 25$d$ for different submerged depths in the lateral plane, (c) $v$ profiles at 25$d$ for different submerged depths in the symmetry plane, (d) $w$ profiles at 25$d$ for different submerged depths in the lateral plane, (e) $uv$ profiles at 25$d$ for different submerged depths in the symmetry plane and (f) -$uw$ profiles at 25$d$ for different submerged depths in the lateral plane.

In addition, higher Reynolds numbers and submerged depth are obtained by quickly opening the water delivery value in the initial stage of water filling, leading to a rapid increase for peak values of Reynolds shear stresses to improve momentum exchange. In summary, the study should be able to effectively reduce the strong impact of side orifice jets on the lock gates, lock walls, and ships moored inside the chamber, resulting in the fact that the phenomena of the gate vibrations and ship collisions with the lock wall are significantly suppressed so as to ensure the structure safeties of ship locks.

## Supporting information

**S1 File. Nomenclature.**
(DOCX)

**S1 Table. Partial statistical data of side orifices longitudinal culvert filling and emptying systems.**
(DOCX)

## Author contributions

**Conceptualization:** Xingxing Zhang.

**Data curation:** Xingxing Zhang.

**Formal analysis:** Xingxing Zhang.

**Funding acquisition:** Xingxing Zhang.

**Investigation:** Han Liu.

**Methodology:** Xingxing Zhang.

**Resources:** Xingxing Zhang.

**Validation:** Han Liu.

**Visualization:** Han Liu.

**Writing – original draft:** Xingxing Zhang, Han Liu.

**Writing – review & editing:** Xingxing Zhang.

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
