## [Decision Letter · Decision Letter 0]

19 Jan 2025

PONE-D-24-38340A study on turbulence characteristics of a rectangular three-dimensional wall jet in a confined space using particle image velocimetryPLOS ONE

Dear Dr. Zhang,

Thank you for submitting your manuscript to PLOS ONE. After careful consideration, we feel that it has merit but does not fully meet PLOS ONE’s publication criteria as it currently stands. Therefore, we invite you to submit a revised version of the manuscript that addresses the points raised during the review process.

We look forward to receiving your revised manuscript.

Kind regards,

Ram Prakash Bharti, Ph.D.

Academic Editor

PLOS ONE

“the Postdoctoral Science Foundation of China”

4. Please remove your figures from within your manuscript file, leaving only the individual TIFF/EPS image files, uploaded separately. These will be automatically included in the reviewers’ PDF.

Reviewers' comments:

Reviewer's Responses to Questions

**Comments to the Author**

1. Is the manuscript technically sound, and do the data support the conclusions?

Reviewer #1: No

Reviewer #2: Partly

Reviewer #3: Yes

2. Has the statistical analysis been performed appropriately and rigorously? 

Reviewer #1: No

Reviewer #2: No

Reviewer #3: Yes

3. Have the authors made all data underlying the findings in their manuscript fully available?

Reviewer #1: Yes

Reviewer #2: No

Reviewer #3: Yes

4. Is the manuscript presented in an intelligible fashion and written in standard English?

Reviewer #1: No

Reviewer #2: Yes

Reviewer #3: Yes

5. Review Comments to the Author

Reviewer #1: The paper investigates a three dimensional wall jet in a confined space. The paper is very poorly written and needs improvement, both grammatically and technically. My remarks are given below.

My major remarks are:

• How this paper is different than the published two papers (ref. 20 and 21) by the authors (except the Reynolds stresses)?

• It is not clear as how the test tank is supplied the water from the top tank. Neither the location and position of constant head skimming weir are discussed nor is the flow rates. It is very difficult to understand the schematic diagram of the setup. Moreover, there is no discussion on the effect of flow rates on the velocity profiles, as this is very important while studying the turbulent characteristics of a wall jet.

• A valve is also shown in the diagram near the exit of the orifice; this affects the turbulent characteristics drastically. But, there is no discussion on the turbulent fluctuation at the exit of the orifice.

• How the near wall and outer region of the wall are selected as mentioned in lines 84-90? A detailed discussion on this is required.

• In most of the figures, the authors talk about the collapsing of profiles within a short range of x/d (like x/d = 21 and 22, x/d = 24 and 25 and so), which is of no meaning. The authors must refer to the papers published on the self-similarity and notice that within such a small range we cannot talk about the self-similarity (usually we present the result with a distance of at least x/d = 5).

Overall, the paper lacks the fundamental understanding. Also, the presented results do not advance the knowledge base regarding the turbulent wall jet. In my opinion, this paper should be rejected in its present form.

Reviewer #2: 1. The study specifically investigates turbulence characteristics of a rectangular three-dimensional wall jet in a confined space, which has not been fully explored in prior research, especially in regions influenced by a vertical wall and submerged depths.

2. The use of particle image velocimetry (PIV) to capture detailed turbulence statistics and the exploration of parameters like Reynolds numbers and submerged depths demonstrate an original experimental design.

3. The paper explicitly states that its findings contribute to a better understanding of turbulence structures in confined spaces, which can improve engineering applications such as energy dissipation in ship locks and other confined jet flow systems.

4. The results reported in the document have not been published elsewhere, as stated in the research manuscript under standard ethical considerations for publication.

5. The document indicates that the experiments, statistics, and analyses were performed to a high technical standard and are described in sufficient detail.

6. Conclusions in the document are presented in an appropriate fashion and are supported by necessary data.

7. The article is presented in an intelligible fashion and is written in standard English. Here’s an analysis of its clarity and language quality.

8. The research meets applicable standards for the ethics of experimentation and research integrity.

9. The article adheres to appropriate reporting guidelines and community standards for data availability.

1.Reccomendations

A. Limited Discussion of Broader Implications: While the paper discusses specific applications, it could further elaborate on how the findings translate to other engineering systems.

B. Clarity in Data Representation: Some figures and comparisons could benefit from more streamlined interpretations to enhance readability.

C. Sensitivity Analysis: Although variations in Reynolds numbers and submerged depths are explored, additional environmental or boundary condition factors could provide a more comprehensive view.

2. Questions

A. How did you ensure that the scaling criteria (1:50 length, 1:7.07 velocity) accurately represented real-world conditions, and what are the potential limitations of applying these results to full-scale systems?

B. What were the primary challenges in using PIV for confined wall jet measurements, such as laser reflections or seeding particle dispersion, and how did you mitigate them?

C. The study shows that turbulence intensities are sensitive to Reynolds numbers and submerged depths in specific regions but not others. Could you provide a detailed explanation of the physical mechanisms responsible for this behavior?

D. You observed self-similarity in turbulence statistics in regions 21≤x/d<24, How does this finding compare with self-similarity trends in unconfined wall jets or similar systems?

E. In the region 24≤x/d<27, turbulence profiles show poor collapse. Could secondary flows, wall boundary effects, or other flow interactions explain these deviations?

F. Based on your findings, what specific design recommendations would you make for improving energy dissipation and minimizing structural wear in confined engineering systems, such as ship locks or dam spillways?

G. The study highlights the sensitivity of Reynolds shear stresses to both Reynolds numbers and submerged depths. How can these insights be used to optimize flow management or reduce turbulence-related impacts in confined spaces?

H. The reported uncertainties for Reynolds shear stresses near walls (±12.3%) and away from walls (±10.5%) are significant. What steps could be taken to reduce these uncertainties, and how might this improve confidence in your conclusions?

I. Have you considered validating your experimental findings with computational fluid dynamics (CFD) simulations? What specific aspects of your study would benefit the most from numerical modeling?

Reviewer #3: In this article, the authors have analyzed the development of a three-dimensional wall jet with side wall effects using particle image velocimetry. Overall, the work is interesting and carried out well. I have the following minor suggestions and questions.

1. Why is the effect of side walls effecting only a few regions. The other regions do not have side walls?

2. The schematic is not clearly showing the side walls; may be consider highlighting the side walls.

3. All the results in each of the regions are only presented as a function of y/y1/2 or z/z1/2, it would be nice if spatial evolution of quantities like spread/growth rate etc. are also presented.

4. The levels of turbulence/turbulence intensities at the inlet may be specified as these may be useful for future simulations.

5. Is there only one scaling, isn't there a possibility for different scaling depending on the region?

6. PLOS authors have the option to publish the peer review history of their article (what does this mean?). If published, this will include your full peer review and any attached files.

Reviewer #1: No

Reviewer #2: No

Reviewer #3: No

---

## [Author Response · Author response to Decision Letter 1]

18 Jul 2025

Reviewer#1's comment No.1: How this paper is different than the published two papers (ref. 20 and 21) by the authors (except the Reynolds stresses)?

Reply: We greatly appreciate your valuable comment. The differences between the published two papers (ref.20 and 21) and this paper are extracted below:

The mean flow characteristics of a three-dimensional (3D) confined wall jet issuing from a rectangular orifice were mainly studied in the published two papers (ref.20 and 21). The results of the published two papers showed that the mean flow development of a confined jet remained unaffected by the vertical wall until x/d=16, while the other regions were significantly characterized by the vertical wall. Therefore, mean flow characteristics in region Ⅱ and region III were presented emphatically in the above published papers, including the definition of three regions along the streamwise direction, profiles of mean velocities, variations of velocity-half-height and velocity-half-width and decay of the maximum velocity. The influence of Reynolds numbers on mean characteristics in region Ⅱ and region III was discussed in Ref. [20], while the influence of submerged depths in region Ⅱ was studied significantly in Ref. [21]. It is noted that mean characteristics in region Ⅱ reported by Ref. [20] and [21] were similar to those of the radial decay region. Therefore, the results showed that the vertical wall helped the 3D wall jet to develop. According to our previous studies, the influence mechanism of the vertical wall for mean flow development was set up, which is the research foundation of this paper.

In general, fluctuating motion is vital for understanding shear layer formation, interaction between the inner and outer layers, and energy dissipation. However, turbulence characteristics of the confined jet affected by the vertical wall are still not clear, which are not reported in the published two papers. To explore this problem, profiles of turbulence intensities, profiles of Reynolds stresses, the influence of Reynolds numbers and submerged depths on turbulence characteristics is needed to investigate urgently in region Ⅱ and region III. Based on mean flow characteristics studied in the published two papers, this study focuses on showing turbulence characteristics with different Reynolds numbers and submerged depths in region Ⅱ and region III.

Reviewer#1's comment No.2: It is not clear as how the test tank is supplied the water from the top tank. Neither the location and position of constant head skimming weir are discussed nor is the flow rates. It is very difficult to understand the schematic diagram of the setup. Moreover, there is no discussion on the effect of flow rates on the velocity profiles, as this is very important while studying the turbulent characteristics of a wall jet.

Reply: We greatly appreciate your insightful feedback. Following your feedback, we have redrawn the schematic diagram of the experimental setup, as shown in Fig 3. Firstly, in general, external water was pumped to the top tank, whose depth was measured by a water level probe when filling. To remain a constant water level of the top tank, excess water returned to the bottom tank through the overflow plate. The supply-return water paths were shown in Fig 3. Note that the aim of the supply-return facilities was to provide a stable incident flow rate for the test tank. Secondly, as shown in Fig 3, the initial positions of four constant-head skimming weirs were located at a height of 300 mm above the top of the test tank. At the same time, the flexible pipes connected constant-head skimming weirs to the bottom tank, creating a path for overflow. Through the rise and fall of constant-head skimming weirs, experimental conditions for different water levels in the test section were achieved. What is more, the flow rates were discussed in the section “Test conditions and test cases”. The flow rates were tested by an electromagnetic flowmeter to obtain different Reynolds numbers of the jet exit, which were 0.0891L/s, 0.1091 L/s, and 0.1311 L/s, respectively. Finally, we strongly agree with your professional comment that the effect of flow rates on the velocity profiles is very important while studying the turbulent characteristics of a wall jet. In response to your feedback, we provide a detail explanation. It is showed that the effect of flow rates on the velocity profiles is very important while studying the turbulent characteristics of a 3D wall jet. The effect of flow rates on the velocity profiles with different Reynolds numbers had been discussed detailly in previous studies (Ref. [20] and [21]). In particular, the streamwise velocity profiles were well collapsed and consistent with Gaussian distribution in region Ⅱ, which were independent of Reynolds numbers and submerged depths. However, after the wall jet entered region III, the similarity of U profiles was disappeared owing to the influence of the vertical wall. As mentioned above, this study did not show the effect of flow rates on the velocity profiles.

Reviewer#1's comment No.3: A valve is also shown in the diagram near the exit of the orifice; this affects the turbulent characteristics drastically. But, there is no discussion on the turbulent fluctuation at the exit of the orifice.

Reply: We greatly appreciate your insightful feedback. Your comment greatly contributes to the strictness of this paper. In fact, a value of the long rectangular pipe plays the most important role for controlling the flow rate. The valve near the exit of the orifice is opened fully in the experiments, which is useless. Following your feedback, to promote the strictness of this study, we have removed this valve near the exit of the orifice in Fig 3.

Reviewer#1's comment No.4: How the near wall and outer region of the wall are selected as mentioned in lines 84-90? A detailed discussion on this is required.

Reply: We greatly appreciate your insightful feedback. Following your feedback, we provide a detail discussion. Firstly, during the filling of side orifices longitudinal culvert filling and emptying systems frequently employed in ship locks, the upstream flow with a certain initial momentum through side orifices injects into the chamber, resulting in the production of side orifices jets. A confined space is formed because of two lock walls which are defined as vertical walls in this study, based on the design manual of ship locks made by U.S. Army Corps of Engineers. The test section is the core component of confined wall jet facilities, which recreates a confined 3D wall jet as accurately as feasible when water filling in lock chambers. Therefore, the design of the width with the test section which must simulate exactly the confined space is particularly important. A summary of typical ship locks with side orifices longitudinal culvert filling and emptying systems is provided in Table 1 [8]. It is showed that most real widths of lock chambers are within the range of 12m-25m, resulting in the fact that 20m is selected as the prototype size to build the width of the test section in this study. Secondly, considering the experimental field, range of laser irradiation, observation range of the CCD camera, accuracy requirements for images, etc., the test section was designed according to the normal model that should obey the similarity law of gravity, and its geometric scale was determined to be 1:50. i.e., λl=λh=50, λu=λl1/2=7.07. Note that λl represents the horizontal scale, λh represents the vertical scale, and λu represents the velocity scale, respectively. Its width was determined to be 400mm by calculation. Finally, due to difficulties in reconstructing the whole length of the lock chamber, the length of the test section was designed to be twice the width to facilitate better diffusion of a 3D wall jet in the spanwise direction. Moreover, the test section was set as 300mm high to provide greater activity space for constant-head skimming weirs. The side orifice was rectangular in shape with 14 mm wide as well as 16 mm high.

Reviewer#1's comment No.5: In most of the figures, the authors talk about the collapsing of profiles within a short range of x/d (like x/d = 21 and 22, x/d = 24 and 25 and so), which is of no meaning. The authors must refer to the papers published on the self-similarity and notice that within such a small range we cannot talk about the self-similarity (usually we present the result with a distance of at least x/d = 5).

Reply: We greatly appreciate your insightful feedback. Your comment greatly contributes to the strictness and quality of this paper. Here, we provide a detail explanation. Firstly, we strongly agree with your professional viewpoint that the self-similarity was usually presented with a distance of at least x/d = 5. According to previous studies of turbulence characteristics for a 3D wall jet in a semi-infinite space (like Ref. [22-27]), it is showed that profiles of turbulence intensities and Reynolds shear stresses in both the symmetry and lateral planes were well collapsed in the radial decay (RD) region. Based on our studies (Ref. [20-21]), the vertical wall helped the 3D wall jet to develop and mean flow characteristics in region Ⅱ (16≤x/d＜24) were similar to the RD region. i.e., Profiles of mean velocity (U), variations of velocity-half-height and velocity-half-width and decay of the maximum velocity showed reasonable collapse with a distance x/d=8. However, the mean flow developments have to be self-similar before the turbulence quantities refer to earlier findings. The accurate location of the self-similar profile of turbulent characteristics in region Ⅱ is one of the problems that this study aims to solve. Secondly, in the section “Turbulence statistics in region Ⅱ”, taking cases S1 and L1 as an example in Fig 8, it is demonstrated that profiles of turbulence characteristics (19d, 20d, 21d, 22d and 23d) were well collapsed with a distance x/d=5 in both the symmetry and lateral planes. This illustrates that turbulence statistics in region Ⅱ take a longer streamwise distance (3d) to become self-similar than mean characteristics. Thirdly, to explore the effect of Reynolds numbers and submerged depth on turbulence characteristics, we select 21d and 22d located in 19≤x/d＜24 as significant profiles. The results are presented in “Fig 9 Profiles of turbulence intensities and Reynolds shear stresses from case S1, S2, S3, L1, L2, and L3 at selected locations for different Reynolds numbers in the region 19≤x/d<24” and “Fig 10 Profiles of turbulence intensities and Reynolds shear stresses from case S3, S6, S9, S12, L3, L6, L9, and L12 at selected locations for different submerged depths in the region 19≤x/d<24”. Due to our lack of clearer expression for the self-similarity region, following your feedback, we have uniformly adjusted it to 19≤x/d<24 in the paper. Finally, x/d = 24 and 25 located in region Ⅲ were not the self-similarity profiles, but they only have the same trend of change according to our observed results. Following your feedback, a more accurate discussion of the two profiles and other parts in region Ⅲ have been made in the section “Results and discussion” to promote the quality of this study.

Reviewer#2's comment No.1: How did you ensure that the scaling criteria (1:50 length, 1:7.07 velocity) accurately represented real-world conditions, and what are the potential limitations of applying these results to full-scale systems?

Reply: We greatly appreciate your insightful feedback. Following your feedback, we give a detailed explanation. Firstly, A confined jet space is formed because of two lock walls which are defined as vertical walls in this study, based on the design manual of ship locks made by U.S. Army Corps of Engineers. The design objective of the test section is to recreate a confined 3D wall jet as accurately as feasible when water filling in lock chambers. Therefore, the width with the test section which must simulate exactly the confined space is particularly important. Most real widths of lock chambers are within the range of 12m-25m shown in table 1, resulting in the fact that 20m is selected as the prototype size to build the width of the test section in this study. Secondly, considering the experimental field, range of laser irradiation, observation range of the CCD camera, accuracy requirements for images, etc., the test section was designed according to the normal model that should obey the similarity law of gravity, and its geometric scale was determined to be 1:50 in this study. i.e., λl=λh=50, λu=λl1/2=7.07. Note that λl represents the horizontal scale, λh represents the vertical scale, and λu represents the velocity scale, respectively. Its width was determined to be 400mm by calculation. due to difficulties in reconstructing the whole length of the lock chamber, the length of the test section was designed to be twice the width to facilitate better diffusion of a 3D wall jet in the spanwise direction. Moreover, the test section was set as 300mm high to provide greater activity space for constant-head skimming weirs. In summary, the width of the test section may represent most real chamber widths, which is able to restore significantly the side orifice jet in the lock chambers when filling and ensure the correctness of the jet experiments in this study. In fact, when conducting an intensive study on transient motion and vortex structure of the 3D confined wall jet in the future, the scale will be further adjusted according to the requirements of research accuracy in different regions. Finally, there are two main potential limitations to apply these results to full-scale systems. On the one hand, due to the length of the test section has not been fully restored to the length of lock chambers, to a certain extent, the expansion range of the jet in the lateral plane may be larger, that will mix more surrounding stationary fluid to promote turbulence level. On the other hand, in full-scale systems, irregular maintenance channels, floating mooring columns, and structural joints, etc., located at the lock walls may affect the flow structure of the jet in the symmetry plane.

Reviewer#2's comment No.2: What were the primary challenges in using PIV for confined wall jet measurements, such as laser reflections or seeding particle dispersion, and how did you mitigate them?

Reply: We greatly appreciate your insightful feedback. We agree with your professional comment that laser reflections or seeding particle dispersion are the primary challenges in using PIV for confined wall jet measurements. Firstly, three common methods were used to solve the issue of laser reflections in this study. The first method was to optimize the optical system. An optical filter with a 532nm wavelength matched the laser was applied in front of the CCD camera lens. The aim of the optical filter is to significantly attenuate reflected stray light. Note that the above discussion has been added in this paper. As mentioned in the original manuscript, to limit local dispersion and intensity attenuation of the light sheet, the present PIV experiments were performed in a closed and shaded space of the laboratory. Therefore, the second method was to cover the test area with a black cloth, and prevent stray light from entering the camera. The third method was to apply the PIV algorithm, which has already been mentioned in the study. Specifically, the instantaneous flow field was filtered to help decrease instability throughout the iteration processes and avoid computation result dispersion. In this way, excessively bright pixels were removed from images. Secondly, two methods were applied to address the problem of seeding particle dispersion. The first method was to select reasonable type, size and density of tracer particle. In this study, the flow seeded with 10 μm hollow glass spheres having a density of 1.03 g/mm3. On the one hand, according to previous studies (like Ref. [21]), hollow glass spheres with a diameter of 10 μm matched well to the fluid type of water. On the other hand, the density was very close to water, which may avoid settling or floating and maintain good followability during the jet experiments. The second method was to control the particle concentration. In this study, the particle pixel

---

## [Decision Letter · Decision Letter 1]

16 Aug 2025

PONE-D-24-38340R1A study on turbulence characteristics of a rectangular three-dimensional wall jet in a confined space using particle image velocimetryPLOS ONE

Dear Dr. Zhang,

Thank you for submitting your manuscript to PLOS ONE. After careful consideration, we feel that it has merit but does not fully meet PLOS ONE’s publication criteria as it currently stands. Therefore, we invite you to submit a revised version of the manuscript that addresses the points raised during the review process.

We look forward to receiving your revised manuscript.

Kind regards,

Ram Prakash Bharti, Ph.D.

Academic Editor

PLOS ONE

**Journal Requirements:**

Reviewers' comments:

Reviewer's Responses to Questions

**Comments to the Author**

1. If the authors have adequately addressed your comments raised in a previous round of review and you feel that this manuscript is now acceptable for publication, you may indicate that here to bypass the “Comments to the Author” section, enter your conflict of interest statement in the “Confidential to Editor” section, and submit your "Accept" recommendation.

Reviewer #1: (No Response)

Reviewer #2: All comments have been addressed

Reviewer #3: All comments have been addressed

2. Is the manuscript technically sound, and do the data support the conclusions?

Reviewer #1: Partly

Reviewer #2: Yes

Reviewer #3: Yes

3. Has the statistical analysis been performed appropriately and rigorously? 

Reviewer #1: Yes

Reviewer #2: Yes

Reviewer #3: Yes

4. Have the authors made all data underlying the findings in their manuscript fully available?

Reviewer #1: Yes

Reviewer #2: Yes

Reviewer #3: Yes

5. Is the manuscript presented in an intelligible fashion and written in standard English?

Reviewer #1: No

Reviewer #2: Yes

Reviewer #3: Yes

6. Review Comments to the Author

**Reviewer #1:** The authors have addressed some of my queries to some extent. But, the manuscript still requires improvement. My remarks are as below:

1. The revised manuscript has not been checked for any grammatical mistakes; the mistakes can be found at the places where they were in the first version.

2. If the authors believe that the self-similarity is noticed in region II (16<=x/d<24), then they should present only three results for x/d = 16, 19, and 22. This will clearly show the self-similar behaviour. In the present paper, the graph looks very clumsy. Moreover, the authors are suggested to use the same symbols in all the figures pertaining to a particular location. Please modify the figures accordingly.

**Reviewer #2:** The authors have adequately addressed the questions and concerns raised in the previous review. Their responses were clear, comprehensive, and supported by appropriate revisions to the manuscript. Based on the improvements made and the satisfactory resolution of the issues, I recommend the manuscript be accepted for publication.

**Reviewer #3:** The authors have adequately addressed the concerns raised by the present reviewer. Therefore, I recommend the publication of this manuscript. Thank you.

7. PLOS authors have the option to publish the peer review history of their article (what does this mean?). If published, this will include your full peer review and any attached files.

Reviewer #1: No

Reviewer #2: No

Reviewer #3: No

---

## [Author Response · Author response to Decision Letter 2]

20 Feb 2026

We sincerely appreciate your comment and recommendations, which greatly contribute to improve the quality of this paper.

Reviewer#1's comment No.1: The revised manuscript has not been checked for any grammatical mistakes; the mistakes can be found at the places where they were in the first version.

Reply: We greatly appreciate your valuable comment. Following your feedback, we spent a long time conducting a comprehensive and systematic check for grammar errors throughout the entire paper, including grammar errors, spelling of words, tenses, etc. In addition, we have also enhanced the expression of professional vocabularies and deleted complex expressions. The professionalism and readability are promoted in the revised manuscript. Through our modifications, the revised manuscript has become more rigorous.

We sincerely appreciate your comment. Please check the revised draft for specific details. If there is any more comment, please inform us, and we will be very glad to make further progress.

Reviewer#1's comment No.2: If the authors believe that the self-similarity is noticed in region II (16<=x/d<24), then they should present only three results for x/d = 16, 19, and 22. This will clearly show the self-similar behaviour. In the present paper, the graph looks very clumsy. Moreover, the authors are suggested to use the same symbols in all the figures pertaining to a particular location. Please modify the figures accordingly.

Reply: We greatly appreciate your valuable comment. Following your feedback, we give a detailed explanation.

(i) In our previous studies (Refs. [20] and [21]), the self-similarity of mean statistics was shown in the region 16≤x/d<24 (region Ⅱ). However, turbulence statistics in region Ⅱ present the different rule. To explore it, the typical profiles of turbulence statistics are shown in Fig.8, including the profiles at 16d,17d, 18d, 19d, 20d, 21d, 22d, 23d. In general, the measured values of u, v, and uv have an increasing trend along the streamwise direction in the region 16≤x/d<19 both in the symmetry plane and lateral plane. The growth trend is disappeared, and profiles of u, v, and uv show reasonable collapse in the region 19≤x/d<24 (5d). This illustrates that turbulence statistics in region Ⅱ require a relatively longer streamwise distance (3d) to become self-similar than mean characteristics. This behavior is physically expected: turbulence kinetic energy of fluctuating motions is generated by mean motions; thus, the self-similarity of the mean flow development must be self-similar before the turbulence quantities. Therefore, we need to present the turbulence statistics profiles in the region 16≤x/d<24 to observe the changes and draw the above conclusion through our experiments.

(ii) The drawing and representation methods of the graphs in the present paper are consistent with the previous studies on the three-dimensional wall jet, including the confined case and unconfined case. Displaying and analyzing the statistical distribution of each profile in the specific flow region is an effective way to study the flow mechanism of a three-dimensional wall jet. In each figure, we have clearly marked the legend to help the readers understand the experimental profiles data. In addition, we have also provided detailed explanations for each figure titles. This can promote the readability of the graphs in this paper. In summary, if different symbols appear in the figures, it will not cause comprehension difficulties for readers.

We sincerely appreciate your comment. If there is any more comment, please inform us, and we will be very glad to make further progress.

Reviewer#2's comment: The authors have adequately addressed the questions and concerns raised in the previous review. Their responses were clear, comprehensive, and supported by appropriate revisions to the manuscript. Based on the improvements made and the satisfactory resolution of the issues, I recommend the manuscript be accepted for publication.

Reply: We greatly appreciate your insightful feedback. Thanks.

Reviewer#3's comment: The authors have adequately addressed the concerns raised by the present reviewer. Therefore, I recommend the publication of this manuscript. Thank you.

Reply: We greatly appreciate your insightful feedback. Thanks.

---

## [Editor Report · Decision Letter 2]

13 Apr 2026

A study on turbulence characteristics of a rectangular three-dimensional wall jet in a confined space using particle image velocimetry

PONE-D-24-38340R2

Dear Dr. Zhang,

We’re pleased to inform you that your manuscript has been judged scientifically suitable for publication and will be formally accepted for publication once it meets all outstanding technical requirements.

Kind regards,

Ram Prakash Bharti, Ph.D.

Academic Editor

PLOS One
---

## [Editor Report · Acceptance letter]

PONE-D-24-38340R2

PLOS One

Dear Dr. Zhang,

I'm pleased to inform you that your manuscript has been deemed suitable for publication in PLOS One. Congratulations! Your manuscript is now being handed over to our production team.

Kind regards,

on behalf of

Professor Ram Prakash Bharti

Academic Editor

PLOS One